# The long noncoding RNA H19 regulates tumor plasticity in neuroendocrine prostate cancer

Neha Singh [1,16], Varune R. Ramnarine[2,3,16], Jin H. Song[1,4], Ritu Pandey[1,4], Sathish K. R. Padi [1,5], Mannan Nouri [2], Virginie Olive[1,6], Maxim Kobelev [2], Koichi Okumura [1,7], David McCarthy[8], Michelle M. Hanna[8], Piali Mukherjee [9], Belinda Sun [4], Benjamin R. Lee[1], J. Brandon Parker [10], Debabrata Chakravarti [10], Noel A. Warfel [1,4], Muhan Zhou[1], Jeremiah J. Bearss[1], Ewan A. Gibb[11], Mohammed Alshalalfa [12], R. Jefferey Karnes[13], Eric J. Small [12], Rahul Aggarwal [12], Felix Feng [12], Yuzhuo Wang [2], Ralph Buttyan[2], Amina Zoubeidi[2], Mark Rubin [14], Martin Gleave[2], Frank J. Slack [3], Elai Davicioni[11], Himisha Beltran [15], Colin Collins [2✉] & Andrew S. Kraft [1,4✉]

Neuroendocrine (NE) prostate cancer (NEPC) is a lethal subtype of castration-resistant prostate cancer (PCa) arising either de novo or from transdifferentiated prostate adeno-carcinoma following androgen deprivation therapy (ADT). Extensive computational analysis has identified a high degree of association between the long noncoding RNA (lncRNA) *H19* and NEPC, with the longest isoform highly expressed in NEPC. *H19* regulates PCa lineage plasticity by driving a bidirectional cell identity of NE phenotype (*H19* overexpression) or luminal phenotype (*H19* knockdown). It contributes to treatment resistance, with the knockdown of *H19* re-sensitizing PCa to ADT. It is also essential for the proliferation and invasion of NEPC. *H19* levels are negatively regulated by androgen signaling via androgen receptor (AR). When androgen is absent SOX2 levels increase, driving *H19* transcription and facilitating transdifferentiation. *H19* facilitates the PRC2 complex in regulating methylation changes at H3K27me3/H3K4me3 histone sites of AR-driven and NEPC-related genes. Additionally, this lncRNA induces alterations in genome-wide DNA methylation on CpG sites, further regulating genes associated with the NEPC phenotype. Our clinical data identify *H19* as a candidate diagnostic marker and predictive marker of NEPC with elevated *H19* levels associated with an increased probability of biochemical recurrence and metastatic disease in patients receiving ADT. Here we report *H19* as an early upstream regulator of cell fate, plasticity, and treatment resistance in NEPC that can reverse/transform cells to a treatable form of PCa once therapeutically deactivated.

A full list of author affiliations appears at the end of the paper.

Neuroendocrine prostate cancer (NEPC) is a highly aggressive and lethal subtype of prostate cancer (PCa) capable of widely metastasizing to organs and bone[1]. Patients with NEPC have limited therapeutic options, and the median overall survival is ~7 months to ~4 years from the time of diagnosis[2,3]. While the disease can develop de novo (dNEPC)[4], it occurs primarily after treatment (tNEPC) arising by a complex process of neuroendocrine transdifferentiation (NEtD) of prostate adenocarcinoma (AdPC). This cellular transformation results from selective pressures from potent androgen receptor (AR) pathway inhibition in castration-resistance prostate cancer (CRPC)[5–8]. With the introduction of highly potent AR-targeting agents, the incidence of tNEPC is increasing[3,9,10]. Manifestations of this subtype include low levels of prostate-specific antigen (PSA) secretion, indifference to AR pathway inhibition, reduced AR protein expression, and the presence of lytic bone lesions and visceral metastasis[11–13]. NEPC can present with either a small or large cell phenotype and is identified histologically by its tumor morphology and expression of neuroendocrine (NE) markers, chromogranin A (CHGA), synaptophysin (SYP), and neuron-specific enolase (NSE)[11,14].

Multiple genetic alterations exist in NEPC, including deletions or mutations in *TP53*, *RB1*, and *PTEN*[15–17], as well as over-expression of *AURKA*, *N-MYC*, and *PEG10*[18,19]. Overexpression is seen in additional drivers/modulators of NEPC, including the polycomb-mediated silencing proteins, e.g., EZH2 and CBX2[20], BRN2[21], ONECUT2[22], DLL3[23], SRRM4[24], and HP1α[25], while the REST protein complex is decreased[26]. In addition, the induction of transcription factors (TFs) SOX2, SOX11, and early stem cell markers such as MYC, OCT4, and KLF4 is involved in NEtD[27]. Despite these discoveries, the mechanism driving NEtD remains elusive and likely involves multiple interacting genetic and epigenetic events. We recently uncovered the landscape of dysregulation in long noncoding RNAs (lncRNAs) in NEPC with *H19*, *LINC00617/TUNAR*, *NKX2-1-AS1*, and *SSTR5-AS1* showing the highest level of expression[28]. Several other lncRNAs, including *PCAT1*, *PCAT19*, *PCA3*, and *PCGEM1*, have also been reported to play important biological roles in PCa[29]. Here, we focus on *H19* due to its elevated expression in clinical NEPC samples[28].

*H19* is predominantly active during fetal development[30], with the highest expression in developing skeletal and smooth muscles. It is encoded by the *H19/IGF2* imprinted gene cluster located on human chromosome 11p15.5. Defined as an oncofetal gene[31], *H19* becomes downregulated during tissue maturation but can be re-expressed in cancer[32]. *H19* has both oncogenic and tumor suppressor functions in multiple cancer types[33]. Within these tumors, it has a regulatory role in a range of biological processes associated with tumor growth, including genome destabilization, hypoxia, epithelial-to-mesenchymal transition (EMT), and mesenchymal-to-epithelial transition (MET)[33,34].

In the present study, we provide insights into the role of *H19* in NEPC while confirming its significant abundance in multiple NEPC clinical cohorts. Experiments demonstrate that *H19* functions as a driver of lineage plasticity in PCa and can induce an NEPC-like phenotype. Knocking down *H19* reverses this process and induces a luminal-like phenotype with increased sensitivity to androgen deprivation therapy (ADT). Here, we demonstrate that *H19* functions as an epigenetic regulator in NEPC by binding to the PRC2 complex, modulating H3K27me3 and H3K4me3 histone marks. *H19* also prompts alterations in genome-wide DNA methylation on CpG sites. Collectively, this remodels chromatin near AR signaling (ARS) and NE genes, regulates the methylation of ARS and NE gene promoters, and consequentially regulates ARS and NE expression. Therefore *H19* plays an essential epigenetic role in NEPC. Our clinical data reveals that *H19* can be used as a diagnostic biomarker for NEPC and is a predictive marker for biochemical recurrence and metastasis in the context of ADT.

## Results

**H19 is highly expressed in NEPC and associated with increasing Gleason grading and ADT.** Previously, we identified *H19* as one of the most highly deregulated lncRNAs in both NEPC and NEtD[28]. To validate this observation and closely correlate *H19* levels with PCa plasticity, we analyzed several additional clinical NEPC cohorts (Table 1) using an updated version of our lncRNA sequencing pipeline (Methods). This included four recently published cohorts (BCCA, WCM2, WCDT, and GRID) and four cohorts (VPC-P, VPC-M, JHMI-N, and WCM1) from our original study. Our pipeline's latest iteration detects 45,031 lncRNAs, of which 40,328 are annotated as such. We quantified 13,764 lncRNAs, 11,886 antisense, and 16,321 pseudogene transcripts (all annotated), within the three major lncRNA subclasses. All cohorts displayed loss of AR-activity (*AR*, *KLK2*, and *PCA3*), upregulation of NEPC bio-markers (*CHGA/B*, *SYP*, *NSE*, *SSTR5*, and *SSTR5-AS1*), dysregulation of NEPC oncogenes/tumor suppressors (*SRRM4*, *PEG10*, *REST*, *EZH2*, and *BRN2*), and dysregulation of NEPC TFs (*SOX2*, *SOX9*, and *SOX11*)—ensuring our pipelines quantifications were accurate (Supplementary Figs. 1–3, Supplementary Data 1–2). Inconsistent patterns or non-significant changes were seen with limited sequencing depth (WCDT) or alternate NEPC pathology (dNEPC in BCCA).

Expression of *H19* was highest in NEPC and was significantly increased in all cohorts compared to control samples (Fig. 1A, B, Supplementary Data 2, $p < 0.05$), except the dNEPC (BCCA) cohort. It is unclear if this resulted from the high tumor cellularity in matched benign samples or disease pathology. In some cohorts (WCM1 and WCDT), *H19* was the topmost differentially expressed lncRNA of all annotated noncoding RNAs across the entire transcriptome. Increasing expression levels of *H19* were associated with increasing Gleason grade and treatment status (Fig. 1A—VPC). *H19* expression was lowest in low-risk (Gleason<7), higher in Gleason>7 ($p < 0.01$) PCa, further increased in neoadjuvant hormone-treated (NHT) patients ($p = 0.0886$), and most highly expressed in NEPC ($p$-value $< 0.01$).

To support the involvement of *H19* in NEPC, we calculated pairwise correlations using Pearson correlation of *H19* with NEPC markers in the WCM1 cohort (Fig. 1C, Supplementary Fig. 4A). Positive correlations were observed between *H19* and classical NEPC markers CHGA, CHGB, and SYP, as well as the NEPC oncogene MYCN ($R = 0.59$, $p = 0.00012$; $R = 0.78$, $p = 1.5e-08$; $R = 0.70$, $p = 1.2e-06$; $R = 0.72$, $p = 5.2e-07$, respectively). Negative correlations were observed between *H19* and AR and SPDEF (Fig. 1C, Supplementary Fig. 4A), both deactivated in NEPC ($R = -0.65$, $p = 1.2e-05$, $R = -0.52$, $p = 0.00087$, respectively). Due to each cohort's rarity and small sample sizes, we amalgamated (see Methods for analysis and normalization) all our sequenced clinical NEPC cohorts. Unsupervised hierarchical clustering of *H19* along with 38 known NEPC genes/lncRNAs within this merged cohort showed distinct separation of AdPC and NEPC samples with no clear cohort bias (Fig. 1E). We next searched for correlations with previously identified TFs and oncogenes of NEPC to identify putative *H19* regulators/targets. Interestingly, we found SOX2 and EZH2 as significantly, positively correlated (Fig. 1C—$R = 0.61$, $p = 7.2e-05$; $R = 0.67$, $p = 4.7e-06$). To investigate how *H19* associates with NE activation and AR deactivation, we compared its expression to available NEPC and AR signature scores in WCM1. We observed a strong positive correlation to NEPC and a negative correlation to AR signature scores (Fig. 1D, $R = 0.75$, $p = 7.2e-08$; $R = -0.73$, $p = 2.8e-07$). To investigate how this result would change across CRPC phenotypes, we repeated our correlation analyses to include WCM2 samples (WCM1 + WCM2) and found highly significant correlations to

**Table 1 Study cohorts and clinical variables.**

| Study Details | | | Pathology | | | | | | | Genomic Footprint | Gleason Grade | | | | Clinical Features and Outcomes | | | | |
|---|---|---|---|---|---|---|---|---|---|---|---|---|---|---|---|---|---|---|---|
| Institute | Profiling technology | Size | BE | AD | CRPC | AD-NHT | MX-P | dNEPC | tNEPC | NEPC | −6 | 7 | 8+ | UN | ADT | RT | BCR | MET | PCSM |
| JHMI | Microarray | 33 | 0 | 13 | 0 | 0 | 10 | 0 | 10 | NA | 0 | 0 | 12 | 1 | NA | NA | NA | NA | NA |
| BCCA | Sequencing | 15 | 5 | 0 | 0 | 0 | 0 | 10 | 0 | NA | NA | NA | NA | NA | NA | NA | NA | NA | NA |
| VPC-P | Sequencing | 75 | 0 | 56 | 0 | 14 | 0 | 0 | 5 | NA | 23 | 4 | 29 | 0 | NA | NA | NA | NA | NA |
| VPC-M | Sequencing | 9 | 0 | 4 | 0 | 0 | 0 | 0 | 5 | NA | NA | NA | NA | NA | NA | NA | NA | NA | NA |
| WCM1 | Sequencing | 37 | 0 | 30 | 0 | 0 | 0 | 0 | 7 | NA | 2 | 23 | 5 | 0 | NA | NA | NA | NA | NA |
| WCM2 | Sequencing | 49 | 0 | 0 | 34 | 0 | 0 | 0 | 15 | NA | NA | NA | NA | NA | NA | NA | NA | NA | NA |
| WCDT | Sequencing | 45 | 0 | 0 | 37 | 0 | 0 | 0 | 8 | NA | NA | NA | NA | NA | N | NA | NA | NA | NA |
| **SUBTOTAL** | | 263 | 5 | 103 | 71 | 14 | 10 | 10 | 50 | NA | 25 | 27 | 46 | 1 | N | NA | NA | NA | NA |
| GRID-BX | Microarray | 9439 | 0 | 9439 | 0 | 0 | 0 | 0 | 0 | 86 | 3127 | 4917 | 1393 | 2 | NA | NA | NA | NA | NA |
| GRID-RP | Microarray | 16806 | 0 | 16806 | 0 | 0 | 0 | 0 | 0 | 177 | 1102 | 12121 | 3565 | 18 | NA | NA | NA | NA | NA |
| MCI | Microarray | 574 | 0 | 574 | 0 | 0 | 0 | 0 | 0 | NA | 64 | 281 | 229 | 0 | 134 | 57 | 407 | 222 | 138 |
| MCII | Microarray | 239 | 0 | 239 | 0 | 0 | 0 | 0 | 0 | 263 | 18 | 120 | 101 | 0 | 79 | 27 | 129 | 80 | 37 |
| **SUBTOTAL** | | 27058 | 0 | 27058 | 0 | 0 | 0 | 0 | 0 | 263 | 4311 | 17439 | 5288 | 20 | 213 | 84 | 536 | 302 | 175 |
| **TOTAL** | | 27321 | 5 | 27161 | 71 | 14 | 10 | 10 | 50 | 263 | 4336 | 17466 | 5334 | 21 | 213 | 84 | 536 | 302 | 175 |

All cohort samples and their associated pathology, Gleason grading, clinical features, and outcomes, including matched benign (BE), adenocarcinoma (AD), neoadjuvant hormonally treated adenocarcinoma (AD NHT), castrated-resistant prostate cancer (CRPC), mixed pathology including adenocarcinoma and small cell (MX-P), neuroendocrine prostate cancer (NEPC), de novo NEPC (dNEPC), treatment-induced (tNEPC), Gleason ≦ 6 (−6), Gleason = 7 (7), Gleason ≧ 8 (8+), unknown Gleason (UN), androgen deprivation therapy (ADT), radiotherapy (RT), biochemical recurrence (BCR), metastasis[34], and prostate cancer-specific mortality (PCSM). It is important to note that all of these cohorts are clinical/patient specimens, except for VPC-M, which are cell line or patient-derived xenograft models of AD, NEtD, and NEPC phenotypes. For NHT samples, only a subset of these (n = 9) was used for differential expression analysis and boxplots of Fig. 1A due to 5 with Gleason grading ≤7 or length of time on NHT <1 month.

individual genes and gene signatures identified previously (Supplementary Fig. 4B–D).

Large-scale testing is required to establish if a dysregulated gene is clinically recurrent (i.e., present in a larger population pool). With the infrequency of NEPC and the scarcity of biospecimens, testing this observation is challenging. However, recently a cohort of 26,245 prospective PCa samples (Decipher GRID), using an NEPC fingerprint of 212 genes (small cell genomic signature—SCGS), identified a subset of patients with transcriptomes analogous to NEPC[35]. With none of these patients having received treatment, we hypothesized these cases were representative of dNEPC or AdPC at high risk for lineage plasticity when exposed to ADT. Therefore, using the SCGS signature and isolating patients within the top and bottom 1% percentile of scores, these patients were classified into NEPC (n = 263) and AdPC phenotypes, respectively. After affirming these samples were molecularly concordant to NEPC (Supplementary Fig. 3), we observed that *H19* had the highest differential expression in NEPC vs. AdPC among these genes (Fig. 1B—GRID). Taken together, this data supports the observation that lncRNA *H19* is associated with increasing Gleason grade, treatment status, and the NEPC phenotype.

**H19 is functionally conserved, and the longest isoform is predominant in NEPC.** To support the potential biological importance of *H19*, we tested its level of conservation across eutherian species. Of the 70 species examined, 47 had DNA alignments and synteny blocks for the human *H19* locus. Performing a multiple sequence alignment (MSA) with these sequences, a high degree of alignment for the region encompassing the five core exons of *H19* (Fig. 2A—right side of MSA, Supplementary Fig. 5A) was observed. Previous studies have suggested that the secondary structure of lncRNAs is conserved, signifying their functional biological role[36]. We modeled *H19* secondary structure and observed a stable minimum free energy (MFE) structure (Supplementary Fig. 6A–D) conserved across our test species. We analyzed a ~1000 nt region spanning the five core exons of *H19* from our MSA and MFE structure (Supplementary Fig. 7) and mostly observed a low covariance (−1 to −2) in the structure across this segment and when analyzing a smaller segment (250 nt) saw a similarly low rate of covariance (Fig. 2B). This data supports that *H19* is not only conserved in its primary sequence but, despite minor sequence differences (Fig. 2B—non-green colored blocks), is also conserved in its secondary structure. Investigating the human form of *H19* more deeply revealed numerous cancer-related SNPs, with ~55% (12/22) being PCa related (Supplementary Data 1) and 43 different TFs, within 23 TF families across 41 different tissue/organs (Supplementary Data 3) capable of binding to its promoter. *H19* has a diverse range of reported functions[33] that could occur due to alternative splicing and usage of specific isoforms in different cellular contexts. *H19* has 13 annotated isoforms (H19i to H19xiii—Fig. 2C). With increased coverage and depth in our VPC cohort, we quantified all isoforms and plotted each in decreasing order based on log2 mean (m) expression (Fig. 2D, Supplementary Fig. 5B). This metric and ranking supported the longest isoform (H19i) as the dominant isoform in tNEPC with >4× fold mean expression across all samples. This result was validated when looking across our other NEPC samples, matching the order from Fig. 2D (Fig. 2E, Supplementary Fig. 5C). Other isoforms of *H19* could be relevant in NEPC, yet their relatively low expression likely results in a non-functional role.

**H19 is elevated in pre-clinical models of NEPC.** To experimentally correlate the level of *H19* with the NEPC phenotype, we evaluated the expression of this lncRNA in NEPC patient-derived

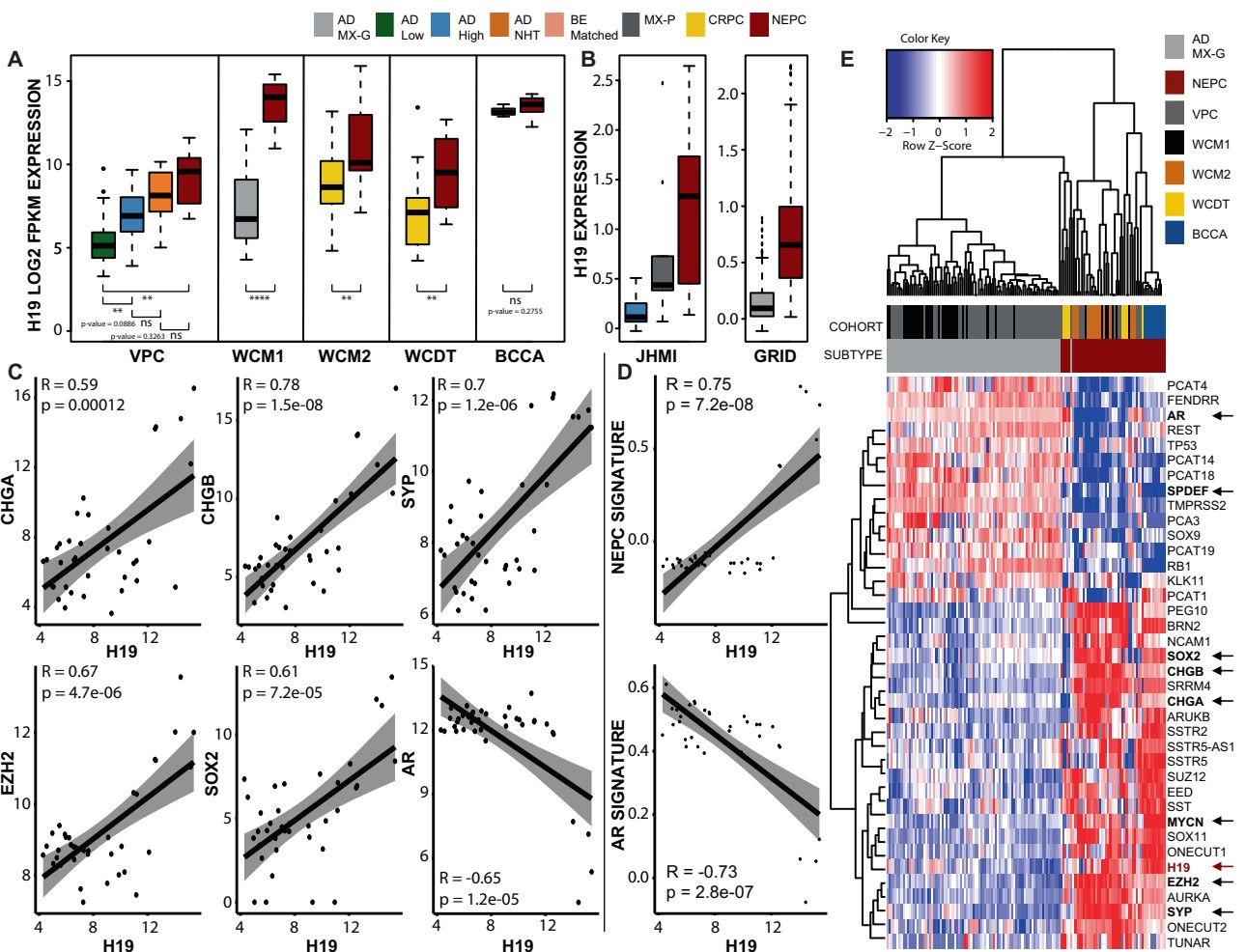

**Fig. 1 H19 expression across clinical NEPC cohorts. A** Sequenced NEPC clinical cohorts. In the VPC cohort ($n = 75$), rising levels of *H19* expression are seen across increasing Gleason grades (Gleason grading $\leqq 6 =$ AD Low and Gleason grading $\geqq 8 =$ AD High), including NHT treated samples and peaks in NEPC. Significant upregulation of *H19* is observed in mixed Gleason grading (MX-G) adenocarcinoma vs. NEPC in the WCM1 cohort ($n = 37$) and CRPC vs. NEPC samples of WCM2 ($n = 49$) and WCDT ($n = 45$) cohorts. Non-significant (ns) yet the elevated expression of *H19* is observed in benign (BE) vs. dNEPC of the BCCA cohort ($n = 15$), yet possibly due to high tumor cellularity in matched BE samples. **B** Microarray NEPC clinical cohorts. Similarly, in the JHMI ($n = 33$) and GRID ($n = 526$) cohorts, rising levels of *H19* from AD High/AD MX-G to mixed AD and small cell pathology (MX-P) to NEPC are observed. Box and Whisker plots display lower quartile, upper quartile, and median bounds of cohort expression at the box's minima, maxima, and centerlines, respectively. Whisker lines display lower (bottom) and upper (top) extreme value ranges. Single data points represent outliers in a cohort. *p* Values were calculated by an unpaired two-sided Student's *t* test. Significance was represented by *$p < 0.05$; **$p < 0.01$; ***$p < 0.001$ and ****$p < 0.0001$ unless specifically noted. **C** In WCM1, *H19* shows a significant positive correlation with CHGA/B, SYP, SOX2, and EZH2 and shows a significant negative correlation with AR expression. **D** Again in WCM1, *H19* shows a significant positive and negative correlation to known NEPC and AR gene signatures, respectively. Correlation coefficients (*R*) and p values (*p*) were calculated using a Pearson correlation statistical test. The shaded area represents confidence intervals at 95%. **E** Unsupervised hierarchical clustering of 38 known genes/lncRNAs in our NEPC ($n = 50$) and AD MX-G ($n = 86$) samples merged across all cohorts show a clear stratification of these two phenotypes. Select genes denoted by arrows have been shown in our correlation analysis from panel **C** and Supplementary Fig. 4A–C.

organoids OWCM-154, -155, -1078, and -1262[37]. *H19* expression was markedly elevated in 3 of the 4 NEPC organoids compared to organoids derived from normal prostate or PCa patients who underwent primary tumor resection for AdPC (Fig. 3A). These samples also exhibited increases in stem cell and NE markers in NEPC vs. non-NEPC samples (Supplementary Fig. 8A). In addition, the NEPC cell lines, NCI-H660 and LASPC-01, compared to AdPC cell lines C4-2B, VCaP, and LNCaP, express elevated *H19*, stem cell genes, and NE markers (Fig. 3B, Supplementary Fig. 8B). Similarly, *H19* is elevated in the tNEPC model cell lines 42D$^{ENZR}$ and 42F$^{ENZR}$ compared to the CRPC cell line V16D$^{CPRC}$ (Supplementary Fig. 8C).

The methylation status of the *H19/IGF2* imprinting control region (ICR1, Supplemental Figs. S8D and S11B) is critical in

driving the expression of this lncRNA. A sensitive bisulfite-free assay (Methylmeter[38]) was used to measure the methylation levels on the CpG site of ICR1 (Supplementary Methods). Endogenously the 1624bp MseI fragment containing the imprinting center CTCF binding regions (1–3) is methylated on the paternal copy. To evaluate DNA methylation changes in NEPC, DNA from cell lines and controls were cleaved with MseI, and the number of methylated versus unmethylated DNA copies in the two fractions were calculated (Supplementary Fig. 8D). The methylation percentage was quantified by coupled abscription PCR signaling (CAPS) (Supplementary Methods, Supplementary Fig. 6D). Results indicate that the AdPC cells (LNCaP, C4-2B, and V16D) have a higher percent of ICR1 methylation when compared to the NEPC cell line (NCI-H660) and NEPC organoid

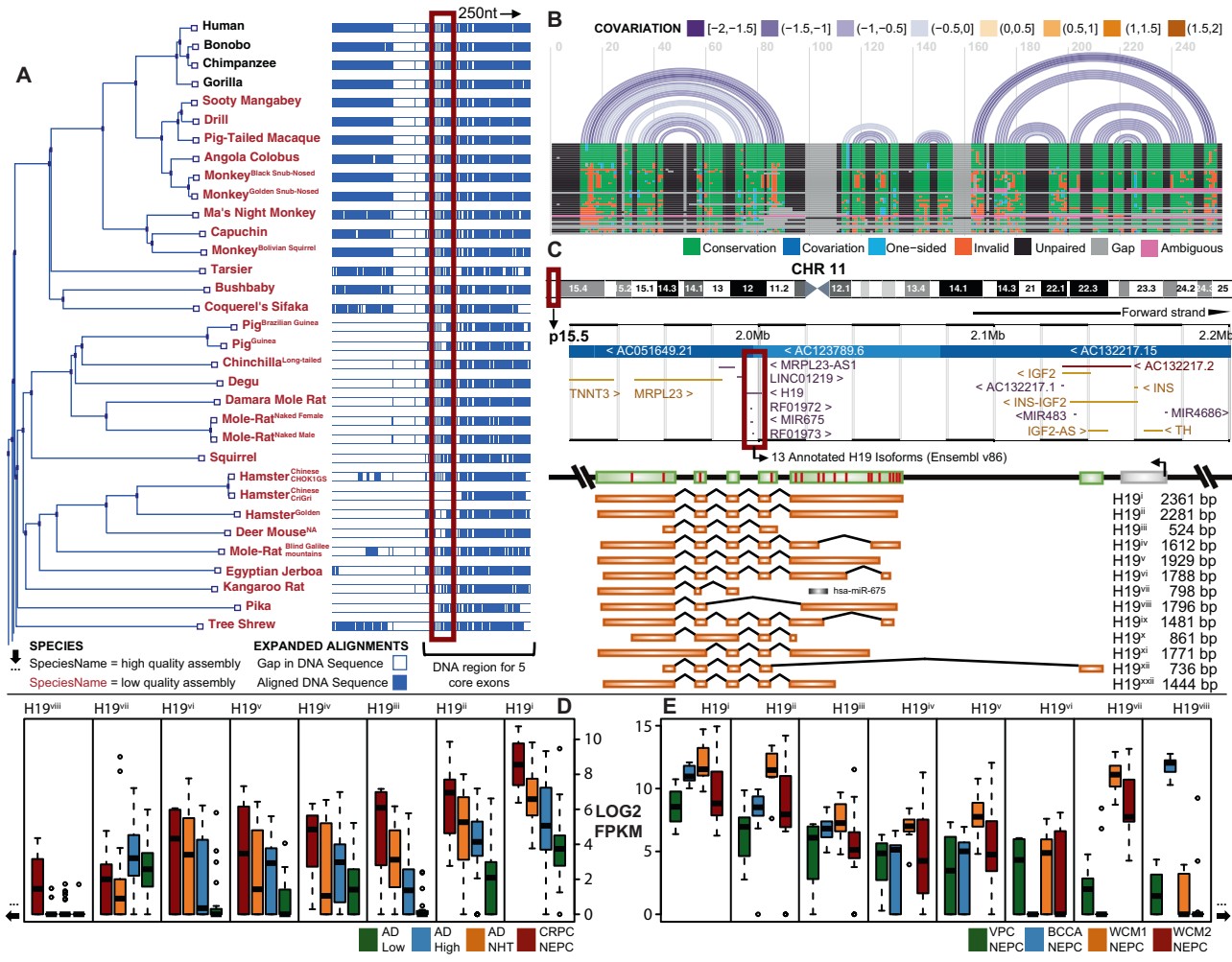

**Fig. 2 H19 Isoform identification and conservation. A** Phylogenetic tree (left side) and multiple sequence alignment (right side) for all available *H19* DNA sequences of eutherian species ($n = 47$) in Ensembl v93. Regions of the MSA that represent gap sequence(s) (white—unaligned sequences) are mostly intronic and regions representing complementary sequence(s) (blue—aligned sequences) are mostly exonic. The five core exons of *H19* (right side of MSA) are predominantly conserved across all test species, suggesting shared functional importance. Due to limited figure space, only 33 of 47 species of this phylogenetic tree have been shown here. The complete ($n = 47$) phylogenetic tree is shown in Supplementary Fig. 5. The highlighted box (red) spans a ~250 nt region where we narrow/zoom into for a **B** detailed MSA (horizontal bars) and integrate RNA secondary structure (purple arcs). Each bar/row represents a test species and is in the order shown from the phylogenetic tree in (**A** and Supplementary Fig. 5). Each arc represents a nucleotide base pairing and binding event. Most of this region is highly conserved (green), yet despite areas with sequence differences (blue, light blue, orange, or gray), these changes do not affect *H19*'s secondary structure, which is stable and contains little covariation (denoted by arc colour). **C** Schematic of human chromosome 11 (top), *H19*'s locus and neighboring genes (middle), and Ensembls' 13 annotated isoforms and exon-exon structures (bottom)—H19i through H19xxii. The order and number of isoforms were assigned based on expression ranking observed in panels (**D**) and (**E**). Each isoform (orange) is transcribed from the DNA region (green) through alternative splicing events (red bars). Each red box in this panel highlights the region expanded below them. **D** Expression of each isoform in various clinical subgroups of the VPC cohort ($n = 75$). H19i (rightmost boxplot) displays the highest expression across all subgroups, H19ii second, and so forth going left. The order of isoforms in this plot was determined by decreasing mean (m) expression across subgroups. **E** Expression of each *H19* isoform in order (left to right) from panel **D** for NEPC samples only in VPC, BCCA, WCM1, and WCM2 cohorts ($n = 40$). Due to limited space, only 8 of 13 isoforms were shown in panels (**D**) and (**E**). Box and Whisker plots display lower quartile, upper quartile, and median bounds of cohort expression at the box's minima, maxima, and centerlines, respectively. Whisker lines display lower (bottom) and upper (top) extreme value ranges. Single data points represent outliers in a cohort. The complete set of *H19* isoform expression plots ranked by decreasing m is shown in Supplementary Fig. 5.

(OWCM-1262), both of which demonstrate methylation in the normal imprinting range of 40–60% (Fig. 3C). Similar results were observed within the ICR1 locus of our LTL331 NEPC xenograft model (Supplementary Fig. 11B). These results suggest that the elevated level of *H19* seen in NEPC could be secondary to changes in methylation of the imprinting center (ICR1) compared to AdPC.

Approximately 70% of NEPC patients harbor mutations or deletions in *TP53* and *RB1*[27]. In mouse models, *Trp53* mutation cooperates with *Rb1* loss to induce Ar[low], Syp[high] NE-like tumors

resistant to orchiectomy-induced androgen deprivation[16]. To evaluate whether the *Trp53/Rb1* knockout modulated *H19* expression, organoids derived from *Trp53*[flox/flox]/*Rb1*[flox/flox] mouse prostates were transduced with lentivirus expressing Cre recombinase (Cre-GFP), generating double-knockout (DKO) organoids (Supplementary Fig. 8E). Organoids transduced with the lentivirus EV-GFP were used as control. These DKO organoids demonstrated enhanced expression of *H19*, markers of the NE phenotype (Chga, Nse, Syp, Brn2, and Ascl1), and stem cell genes (Klf4, Oct4, and Sox2) (Fig. 3D)[16]. In comparison,

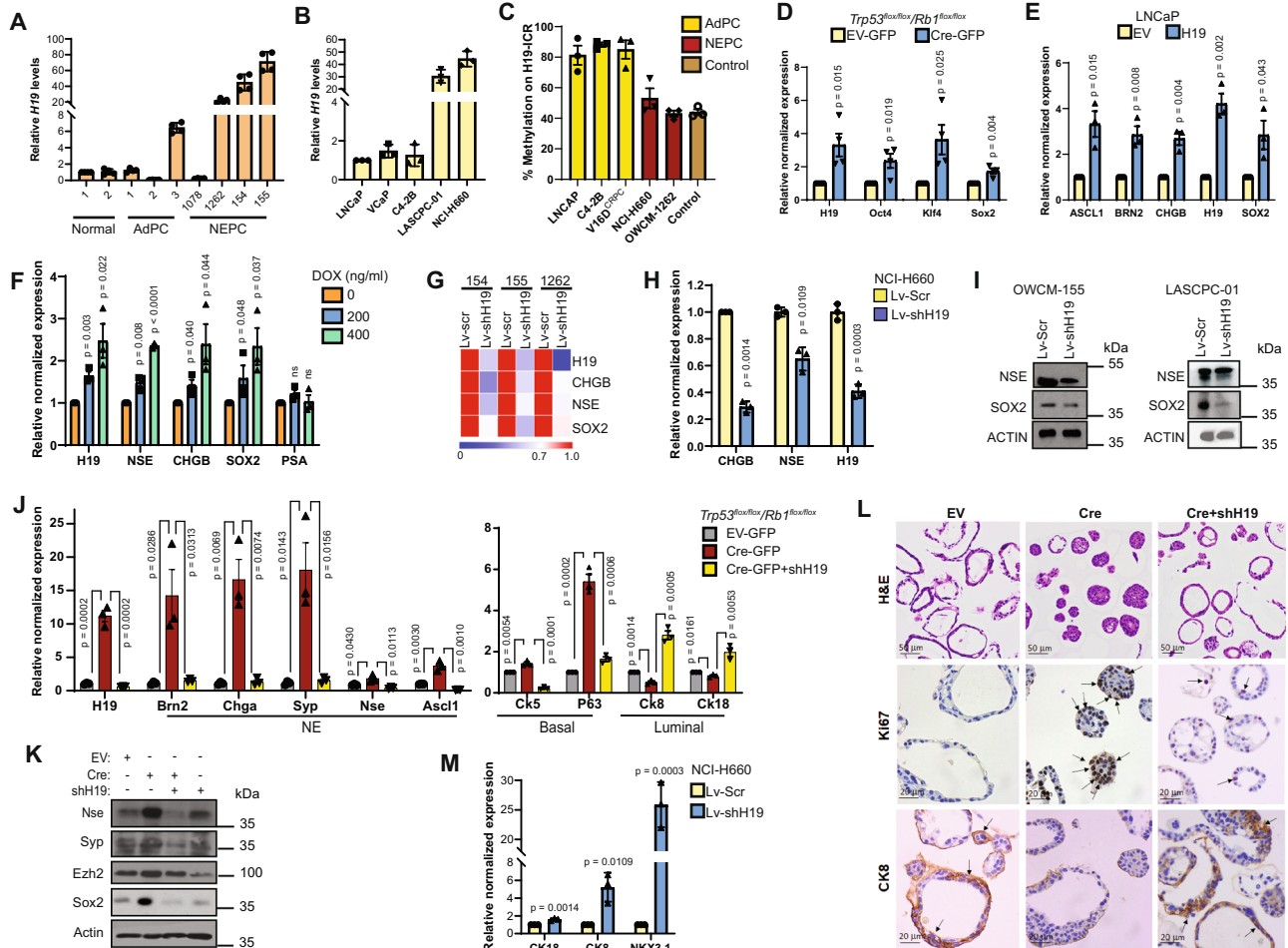

**Fig. 3 H19 is elevated in NEPC organoids and cell lines and controls the expression of stem cell and NE markers.** Relative *H19* expression in **A** patient-derived organoids from normal (1–2) biopsy, AdPC (1–3), and NEPC (OWCM-154, 155, 1078, 1262) and **B** AdPC (LNCaP, C4-2B and VCAP), and NEPC cell lines (LASCPC-01 and NCI-H660). **C** Percent methylation on *IGF2/H19*-Imprinting control region (H19-ICR) derived by Methyl meter assay in representative control (see Supplementary Methods), AdPC, CRPC, and NEPC cell lines. The normal imprinting range was defined as 40–60% methylation. **D** Relative RNA expression of indicated genes after *Trp53/Rb1* DKO induced by lentiviral transduction of Cre-recombinase in *Trp53flox/flox/Rb1flox/flox* mouse prostate organoids (Cre-GFP) vs. control transduced organoids (EV-GFP). **E** Relative RNA expression of *H19* and NE markers in LNCaP overexpressing *H19* vs. control (EV). **F** Relative RNA expression of indicated genes in doxycycline (DOX) inducible H19FL C4-2B cells upon DOX treatment (0, 200, 400 ng/mL; 48 h) (see also Fig. 6F). **G** Heat map of relative RNA expression of indicated genes in control (Lv-Scr) and *H19* knockdown (Lv-shH19) NEPC organoids. **H** Relative RNA expression of indicated genes after *H19* knockdown in NCI-H660 cells (Lv-shH19) vs. control transduced cells (Lv-Scr). **I** Western blot (WB) of SOX2 and NSE in control (Lv-Scr) and *H19* knockdown (Lv-shH19) OWCM-155 and LASCPC-01. **J** Relative RNA expression of representative lineage-specific genes in mouse prostate organoids after *Trp53/Rb1* DKO induced by lentiviral transduction of Cre-recombinase (Cre-GFP) into *Trp53flox/flox/Rb1flox/flox* mouse prostate organoids. *H19* knockdown is denoted as Cre-GFP + shH19. EV transduced organoids (EV-GFP) were used as controls. #p-values < 0.05 vs. EV-GFP organoids. *p-Values < 0.05 vs. Cre-GFP organoids. **K** WB of representative NE genes in the *Trp53flox/flox/Rb1flox/flox* mouse prostate organoids transduced with EV, Cre, Cre+shH19, or shH19. **L** Representative images showing H&E staining (scalebar 50 μm) and immunostaining (scalebar 20 μm) for Ki67 and CK8 in the same organoids from (**J**). Arrows represent areas of intensive staining. **M** Relative RNA expression of luminal markers after *H19* knockdown in NCI-H660 cells (Lv-shH19) vs. control transduced cells (Lv-Scr). Data are mean ± SD (**A, B, H, M**), or mean ± SEM (**C, D, E, F, J**); n = 3 (**B, C, E, F, H, J, M**) or n = 4 (**A, D**) biologically independent replicates. *p* Values were calculated by unpaired two-tailed Student's *t* test.

while single-gene knockout of *Trp53* or *Rb1* in this organoid model had little effect on *H19* or stem cell gene levels. Consistent with these findings, *H19* induction was observed in LNCaP cells after knocking down *TP53* and *RB1* (Supplementary Fig. 8F). Organoids derived from mice with other genetic mutations commonly found associated with NEPC, including *Trp53/Pten* DKO and *Rb1/Pten* DKO, also demonstrated induction of *H19* (Supplementary Fig. 8G, H). Similar results could be seen using a model system in which NEtD is induced by incubating the hormone-dependent cell lines, LAPC-4 and LNCaP, in a stem transition medium that differentiates these cells into an NEPC-like stem-like state[10,39] (Supplementary Fig. 8I). While induction

of NEtD significantly increased *H19* in these cells, levels returned to baseline when the cells were placed back in normal serum. Thus, *H19* is elevated in NEPC in vitro models and is modulated by critical drivers of NEtD.

**H19 regulates the expression of stem cell genes, NE markers, and lineage plasticity.** To examine whether modulating *H19* levels can drive changes in both stem cell and NE genes, we overexpressed *H19* in LNCaP and V16DCPRC cells. This resulted in increased expression of both the NE phenotype (CHGB, BRN2, ASCL1) and stem cell genes (SOX2, NANOG) (Fig. 3E,

Supplementary Fig. 9A). In addition, using a doxycycline (DOX) inducible system for overexpression of *H19*, C4-2B cells demonstrated induction of NE and stem cell markers at both the protein and RNA level (Figs. 3F, 6F-LNCaP).

Stable knockdown of *H19* in both NEPC patient-derived organoids (Fig. 3G, Supplementary Fig. 9B) and NEPC cell lines (NCI-H660, LASPC-01, 42D$^{ENZR}$, and 42F$^{ENZR}$) caused a reduction of NE markers (NSE, CHGB, SYP) and stem cell genes (SOX2, OCT4, NANOG) (Fig. 3G, H, Supplementary Fig. 9C–E). These results were confirmed at the protein level in NEPC organoid OWCM-155 and LASPC-01 cells (Fig. 3I). In 42D$^{ENZR}$ cells with stable *H19* knockdown, overexpression of murine *H19* rescued the expression of NSE and CHGB (Supplementary Fig. 9I, J). Loss of *Trp53/Rb1* in a murine prostate organoid model resulted in the induction of *H19* levels (Fig. 3J). These murine organoids were further transduced with lentivirus encoding shRNA to cause *H19* knockdown (Cre-GFP + shH19). The *H19* knockdown decreased stem cell and NE markers (Fig. 3K) while upregulating the luminal phenotype markers CK8 and CK18. This was confirmed by immunohistochemical CK8 staining (Fig. 3L) and suggested a lineage switch from a NE to luminal (Fig. 3J, L) phenotype. A similar lineage reversal was observed in NCI-H660 and LASPC-01 after *H19* knockdown (Fig. 3M, Supplementary Fig. 9C, S9F). These results demonstrate that PCa lineage plasticity can be reversible, highlighting the potential for bidirectional changes of NEPC.

To examine whether stem cell genes might regulate *H19* levels in NEPC, *SOX2*, *NANOG*, and *OCT4* were individually knocked down in LNCaP cells with previously deleted *TP53/RB1*. Results demonstrated that decreasing each stem cell gene caused a significant reduction in *H19* expression (Supplementary Fig. 9G). Furthermore, as demonstrated previously[40,41], the knockdown of each stem cell gene caused changes in mRNA level of the other two genes investigated (Supplementary Fig. 9G). This finding suggests a possible feed-forward mechanism that controls the levels of this lncRNA within the cell.

**H19 knockdown reduces cell proliferation, invasion, and re-sensitizes resistant cells to enzalutamide.** NEPC is characterized by highly proliferative cells with increased metastatic potential[42]. Stable *H19* knockdown inhibited OWCM-155 proliferation in vitro (Fig. 4A). Similarly, *Trp53/Rb1* DKO murine organoids transduced with shH19 showed markedly suppressed growth ($p = 7 \times 10^{-9}$) (Fig. 4B, C). Similar growth suppression was seen with the knockdown of *H19* in LNCaP-SL and 42F$^{ENZR}$ cells (Supplementary Fig. 10A, B). Furthermore, subcutaneous injection of organoids with *H19* knockdown (OWCM-155-shH19) in mice demonstrated significantly slower growth with reduced tumor weight and volume as compared to mice injected with control (OWCM-155-shSCR) organoids (Fig. 4D, E, Supplementary Fig. 10C). In addition, tumors containing *H19* knockdown were qPCR validated with reduced *H19* and NE markers (Fig. 4F). These results indicate that *H19* is critical both for NEPC growth and differentiation. To examine the ability of *H19* in modulating tumor cell invasion, dissociated mouse *Trp53/Rb1* DKO organoid cells were placed in a transwell with Flourblock inserts (Fig. 4G), and *H19* knockdown was shown to inhibit invasion. Similar results were seen in LNCaP-SL cells (Supplementary Fig. 10D). Together these data confirm that a decrease in *H19* affects the growth and invasive potential of NEPC.

Knockdown of TP53/RB1 in LNCaP cells allows them to become NEPC-like, showing less sensitivity to growth inhibition by the AR antagonist enzalutamide (ENZA)[16]. LNCaP cells containing shTP53/Rb1 are resistant to ENZA-induced growth blockade and instead undergo NEtD. Interestingly, when these

cells are transduced with shH19, they regained their sensitivity to hormone blockade (Fig. 4H) and demonstrate growth inhibition by ENZA. Western blots demonstrated that *H19* knockdown (shH19-C and shH19-D, Supplementary Fig. 9H) abrogated this ENZA induced NEtD, reducing the protein levels of NE markers (Fig. 4I). Conversely, ENZA (2 μM or 5 μM) treatment for 5 days inhibited the growth of control LNCaP cells, but not those with stable overexpression of *H19* (Fig. 4J, K). However, we did observe reduced AR levels with ENZA treatment in *H19* overexpression LNCaP cells, indicating the complexity of these molecular pathways. Together these data highlight the importance of *H19* in regulating the sensitivity of PCa cells to ADT.

**Androgen deprivation and stem cell genes regulate H19 transcription during NEtD.** NEtD is a complex process involving suppressed androgen signaling followed by a stem cell state that allows the cells to dedifferentiate NE phenotype. Using existing RNA-sequencing and microarray expression from the LTL331/LTL331R PdX models of NEtD[19], we sought to identify when H19 is expressed. Analysis of RNA collected during different phases of NEtD (AdPC, post-castration, and NEPC) (Supplementary Fig. 11A) revealed that *H19* transcription is elevated in a biphasic manner, the first increase occurring post-castration and a second during the terminal differentiation to NEPC (Fig. 5A). In this model, the induction of SOX2 occurs primarily in the second phase of NEtD (Supplementary Fig. 11E).

Since *H19* levels increased post-castration, further analysis of the relationship between *H19* and the AR was undertaken. Our computational analysis revealed an inverse correlation between *H19* and *AR* expression (Supplementary Fig. 12A). Consistent with this observation, knockdown or overexpression of *H19* in NCI-660, LASPC-01, and V16D$^{CPRC}$, showed a strong inverse correlation between the PSA (RNA) and *H19* (Supplementary Fig. 12C). To test the direct effects of AR signaling, C4-2B cells were treated with an AR agonist dihydrotestosterone (DHT) or an antagonist, ENZA. The addition of DHT suppressed *H19* expression (Fig. 5B), while ENZA increased *H19* levels (Fig. 5C). Moreover, long-term androgen-deprived LNCaP cells that became neuronal-like and expressed increased NE markers, demonstrated elevated *H19* levels (Fig. 5D). To study the mechanism by which AR regulated *H19*, chromatin immunoprecipitation (ChIP) was performed with the anti-AR antibody on C4-2B cells treated with DHT or vehicle. ChIP-qPCR of the *H19* upstream region demonstrated three ARE binding sites (529, 860, 2284 bp) upstream of the *H19* transcription start site (TSS) (Fig. 5E). These binding sites were significantly enriched for AR binding after DHT treatment (Supplementary Fig. 12D), whereas loss of AR occupancy was demonstrated upon ENZA treatment (Fig. 5F). These findings were validated using KLK3 as a positive control (Fig. 5F, Supplementary Fig. 12D). Using a construct with *H19* proximal promoter (H19-PP, 850 bp) driving a luciferase reporter, experiments further demonstrated that in C4-2B cells, DHT was capable of decreasing reporter activity. Conversely, ENZA treatment increased *H19* transcription, thus confirming that the proximal promoter region is involved in AR regulation of *H19* transcription (Fig. 5G).

Androgen deprivation has been shown to elevate the levels of SOX2[16]. This TF regulates lineage plasticity and the induction of NEtD, suggesting that it could also play a role in regulating *H19* levels. Experiments demonstrated that SOX2 overexpression in LNCaP cells increased *H19* expression while knockdown of SOX2 in C4-2B cells decreased *H19* transcription (Fig. 5H). Likewise, ENZA treatment of C4-2B cells increased both SOX2 and *H19* expression (Supplementary Fig. 12E). To examine the relationship between AR signaling, SOX2, and *H19*, ChIP was carried out with anti-SOX2 antibody on C4-2B cells treated with ENZA

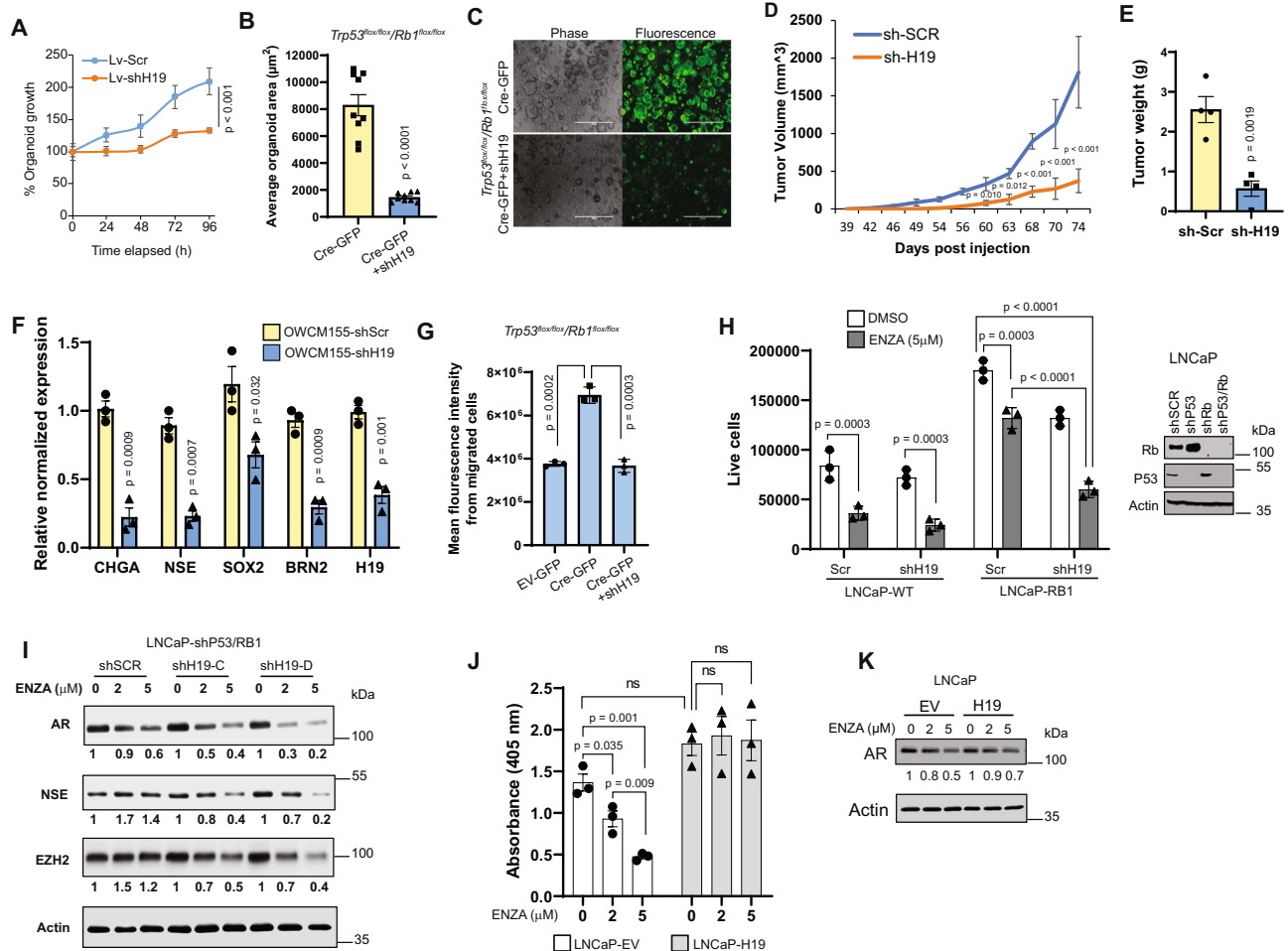

**Fig. 4 Elevated H19 is essential for proliferation and invasion of NEPC-like cells and sensitizes to Enzalutamide (ENZA) growth inhibition. A** OWCM-155 NEPC organoid cells (5000/well) were plated with or without stable *H19* knockdown (Lv-shH19 or Lv-Scr). Line graphs demonstrate the quantification of organoid growth. **B** *Trp53*^flox/flox^/*Rb1*^flox/flox^ mouse organoid cells (5000/well/condition) were plated. Bar plot represents organoid area quantification ($n = 10$; biological replicates) **C** Representative fluorescence images of organoids in Fig. 4B. Scale bar: 1000 μm. **D** Time course of OWCM-155 control (shScr) and *H19* KD (shH19) tumor xenograft growth in $n = 5$ NSG mice. **E** Mean tumor weight of the tumor xenografts ($n = 4$) harvested at 11-week post-injection. **F** Relative mRNA expression of *H19* and NE markers in OWCM-155 shH19 vs. OWCM-155 shScr xenografts. Each bar represents pooled data from tumors from three different mice per group. **G** Organoid invasion assay. Transwell assay using FluoroBlock inserts with 50,000 cells/well of *Trp53*^flox/flox^/*Rb1*^flox/flox^ organoids transduced with EV-GFP, Cre-GFP, and Cre-GFP + shH19 ($n = 3$). Quantification by fluorescence intensity of the migrated cells on the bottom of the transwell, five days post-plating. EV-GFP was used as a control. **H** Growth response (left) of LNCaP cells with WT and *P53/RB1* knockdown (shP53/RB1) with and without *H19* knockdown (shH19 vs. Scr) treated with ENZA (5 μM) for 5 days. DMSO was used for the control treatment. WB (right) of these cells shows the RB and P53 knockout. **I** WB of LNCaP shP53/Rb1 cells with and without two different shH19 (shH19-C, shH19-D) treated with ENZA (2, 5 μM). DMSO was used as a control. **J**, Growth response of LNCaP cells overexpressing *H19* vs. empty vector (EV) treated with ENZA (2, 5 μM) for 5 days. DMSO was a control treatment at 0 μM. **K** Western blot of LNCaP cells used in (**J**) treated with ENZA (2, 5 μM; 72 h). In **H**, **I**, and **K**, WB images were quantified using ImageJ (Methods) with Actin as a control. Data are mean ± SD (**D**, **G**, **H**), or mean ± SEM (**A**, **B**, **E**, **F**, **J**); $n = 3$ (**F**–**H**, **J**) biologically independent replicates. *p* Values were calculated by unpaired two-tailed Student's *t* test (**A**, **B**, **D**–**G**, **J**) or Tukey's multiple comparisons test (**h**).

(10 μM) or vehicle (EtOH) for six days to induce NEtD. Chip-qPCR results demonstrated significant enrichment of SOX2 binding on 2 of the putative SOX2 sites (239, 1563 upstream of *H19* TSS), one of them was found close to the *H19* TSS (Fig. 5I). Furthermore, in 42D^ENZR^ cells, ENZA treatment induced an increase in luciferase activity upon transfecting the wild type H19-PP luciferase construct (850 bp), which was blocked by mutating the SOX2 binding site close to the *H19* TSS (Fig. 5J). This result confirms the ENZA-mediated SOX2 regulation of *H19* transcriptional activity. Together, these experiments point to a mechanism in where androgen signaling initially suppresses *H19* transcription, ADT alleviates this due to augmented SOX2 levels, and therefore *H19* transcription is further elevated (Fig. 5K).

**H19 induces epigenetic changes including modifying histone methylation by binding to the PRC2 complex.** The subcellular localization of a lncRNA can guide the identification of function. We observed a significant level of nuclear expression of *H19* relative to the cytoplasm (Fig. 6A), suggesting that in NEPC, *H19* might predominantly function to regulate gene transcription. Epigenetic reprogramming has been implicated in the NEPC development[20,43]. Our experiments demonstrated that the level of H3K27me3, the target of the PRC2 complex, and H3K4me3 was elevated in NEPC (Fig. 6B) compared to AdPC. The transfection of *H19* into LNCaP and V16D^CRPC^ induced H3K27 and H3K4 trimethylation (Fig. 6C). In addition, the transduction of the NEPC organoid OWCM-155 with Lv-shH19 markedly decreased

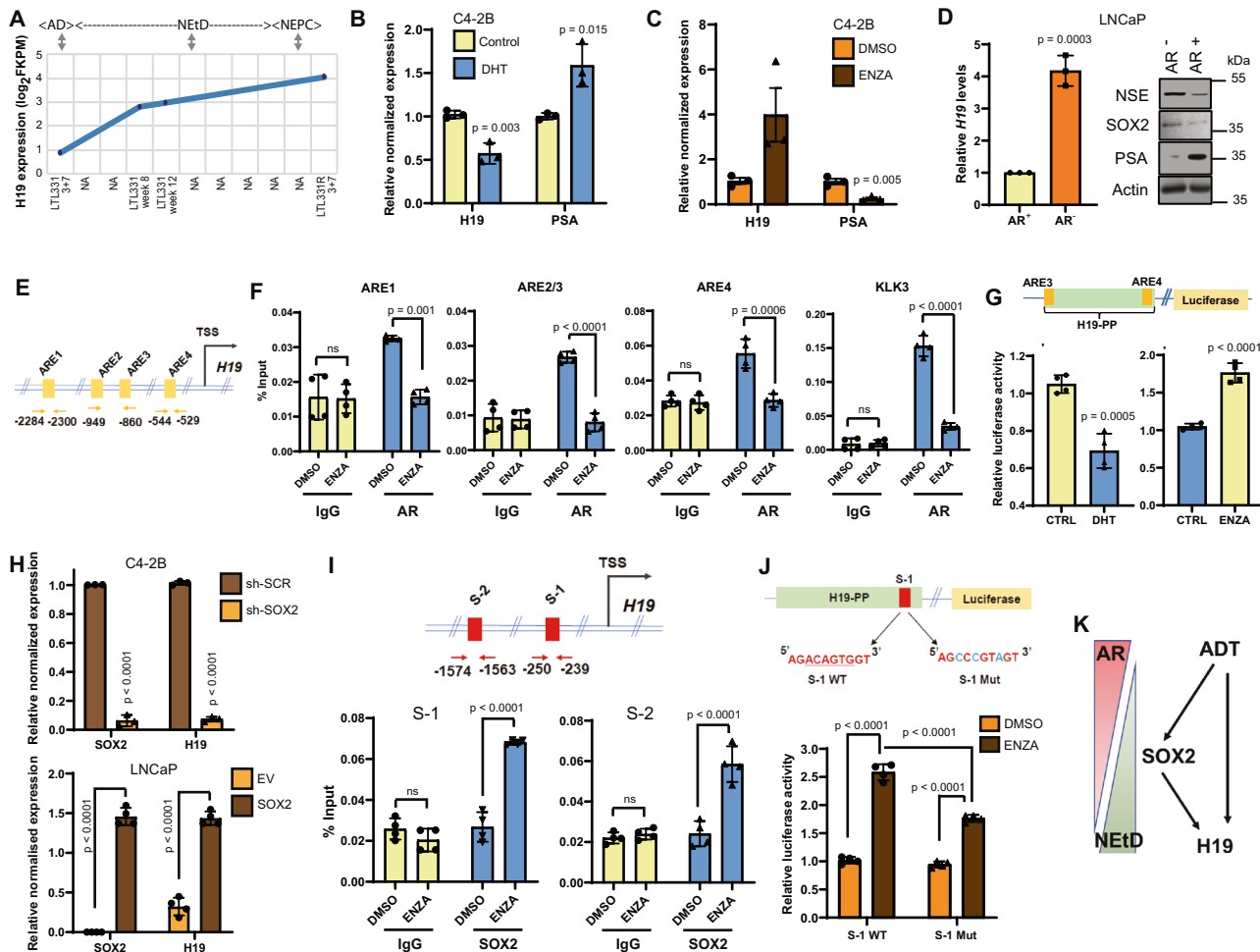

**Fig. 5 Androgen deprivation causes direct binding of SOX2 on the *H19* promoter upregulating its transcription during NEtD. A** *H19* RNA expression during different phases of NEtD in the LTL331 system (*n* = 1). **B** Relative RNA expression of *H19* and PSA in C4-2B cells after DHT (10 ng, 48 h) treatment vs. control (EtOH). **C** Same as **B** after ENZA (10 μM, 5 days) treatment vs. DMSO. **D** Relative RNA expression of *H19* in androgen starved (AR−) LNCaP cells placed in phenol-free media with 10% CSS for 40 days. LNCaP cells grown in regular media (AR+) are controls. The right panel demonstrates WB assay for indicated proteins and Actin (control) using cells in the left. **E** Genomic location of AR binding motifs (ARE) on *H19* promoter. **F** In C4-2B cells vs. control (DMSO) cells, ChIP-qPCR showed AR binding on *H19* promoter and a decrease with ENZA (10 μM) treatment. The KLK3 promoter served as a positive control. **G** Schematic showing luciferase reporter construct with the proximal promoter of *H19* (H19-PP) harboring ARE sites (ARE 3–4) (top). Luciferase reporter activity of the H19-PP in DHT (10 nM) stimulated cells vs. ethanol (control) treatment. The right panel is the same, except for ENZA (10 μM) treatment of C4-2B cells. **H** Relative RNA expression of SOX2 and *H19* in C4-2B cells stably expressing sh-SOX2 vs. sh-SCR control (top); and stably expressing SOX2 vs. empty vector control transduced cells (bottom). **I** Genomic location of SOX2 binding motifs (S-1, S-2) on the *H19* promoter. Bar plots demonstrating ChIP-qPCR are showing enrichment of SOX2 on the *H19* promoter upon ENZA (10 μM) and DMSO (control) treatment in C4-2B cells (bottom). **J** Schematic showing luciferase reporter constructs with H19-PP wild-type (S-1 WT) or mutated (S-1 Mut) SOX2 binding site (mutation in blue) (top). Bar plots showing luciferase reporter activity of H19-PP WT-S1 or Mut-S1 in ENZA (10 μM) treated 42D^ENZR cells (bottom). **K** Diagram depicting the regulation of *H19* during NEtD mediated by androgen deprivation (ADT) and consequently via SOX2. Data are mean ± SD (**B, D, F–J**), or mean ± SEM (**C**); *n* = 3 (**B, C, D, H**) or *n* = 4 (**F, I, J**) biological replicates. *p* Values were calculated by unpaired two-tailed Student's *t* test (**B–D, G–I**) or two-way ANOVA multiple comparisons test (**F, J**).

H3K27me3 while only slightly reducing EZH2 expression (Fig. 6D). Together these data indicate that *H19* plays a role in the epigenetic changes induced during NEtD. *H19* overexpression in LNCaP was shown not to alter the expression levels of PRC2 complex proteins, EZH2, SUZ12, and AEBP5 (data not shown), whereas RNA immunoprecipitation (RIP) in LASCPC-01 (Fig. 6E) and NCI-H660 (Supplementary Fig. 13A) demonstrated the binding of EZH2 to *H19*. RIP analysis of LNCaP and V16D^CRPC cells overexpressing *H19* confirmed that the *H19* transcript was enriched in the immunoprecipitation of endogenous EZH2 (middle) and SUZ12 (right) (Fig. 6E). It has been reported that the association of *H19* with EZH2 at a specific region in the 5′ end of the lncRNA is responsible for PRC2

activity[44]. To test whether this interaction is essential for NEtD, we created an EZH2 binding site deletion fragment (H19^DEL, Supplementary Fig. 13B), with a 5′ deletion, and cloned it into a DOX inducible Tripz vector. The addition of DOX to these cells induced NE marker expression in H19^FL but not in the H19^DEL transfected LNCaP cells (Fig. 6F, Supplementary Fig. 13C, D). These results establish the functional importance of H19/EZH2 binding in mediating NEtD.

To determine the functional epigenetic landscape of changes induced by *H19*, ChIP-sequencing was performed on H3K27me3 and H3K4me3 in V16D^CRPC transduced with *H19* (to induce NEtD) versus V16D^CRPC control cells. We observed significant differential binding of H3K27me3 and H3K4me3 in V16D/H19

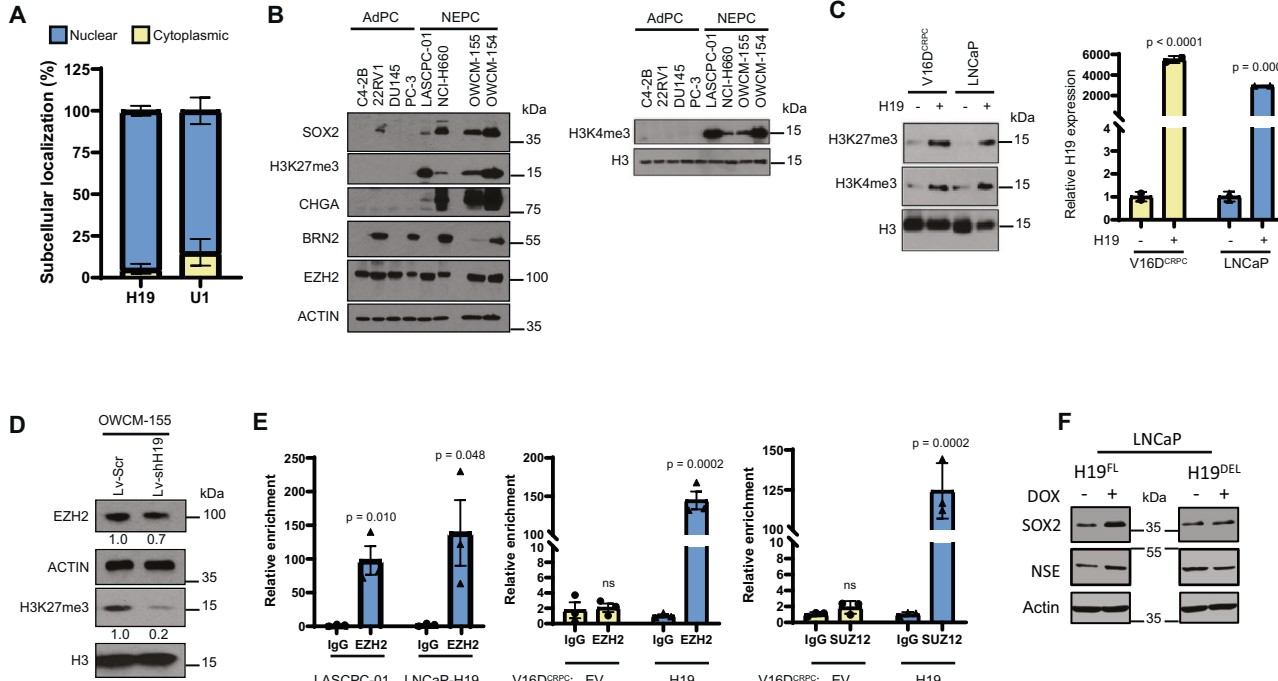

**Fig. 6 H19 induces epigenetic changes including modifying histone methylation by binding to the PRC2 complex. A** Nuclear localization of *H19* in NCI-H660. *y*-Axis represents the percent abundance of RNA. Nuclear U1 RNA was used as a control. **B** WB of NE associated genes (SOX2, CHGA, BRN2, and EZH2), H3K27me3, and H3K4me3 in various AdPC and NEPC cell lines and organoids. **C** WB of H3K27me3, H3K4me3 in CRPC cell line V16D^CRPC and AdPC cell line LNCaP after transient overexpression of *H19*. The bar graph shows the relative *H19* RNA levels in both the cell lines upon *H19* overexpression. 18S was used as an endogenous control. **D** WB analysis of the levels of EZH2 (Actin as control) and histone H3K27me3 level (Histone H3 as control) in Control (Lv-Scr) and *H19* knockdown (Lv-shH19) OWCM-155 NEPC organoids. Numerical values shown under the blot are calculated relative to the control samples. **E** Relative enrichment of *H19* binding to PRC2 complex members EZH2, SUZ12 in LASCPC-01, LNCaP, and V16D^CRPC cells with transient overexpression of control (EV) and *H19* (H19). **F** WB analysis of LNCaP cells stably expressing doxycycline (DOX) inducible H19^FL (full-length *H19*) or H19^DEL (5′ deleted *H19* fragment) with or without DOX treatment (200 ng/mL, 48 h). Actin was used as a control. Data are mean ± SD (**A**, **C**), or mean ± SEM (**E**); n = 3 (**A**, **C**, **E**) biologically independent replicates. *p* Values were calculated by unpaired two-tailed Student's *t* test.

cells as compared to V16D/CTL cells (Fig. 7A, Supplementary Data 13), with a significantly increased binding distribution in proximal promoter region <1 kb from TSS (H3K27me3 = 84%, H3K4me3 = 46%) (Fig. 7B) in cells transduced with *H19*. Upon *H19* overexpression, peak occupancy of H3K4me3 and H3K27me3 was altered in genes that constitute the NEPC signature[35,45,46] and AR signaling genes (Supplementary Data 7–10). Gene ontology (GO) analysis of V16D/H19 cells compared to control demonstrated that neuronal regulatory pathways had significant changes in H3K27me3 levels while H3K4me3 changes were found in cytoskeletal and neuronal regulatory pathways (Fig. 7C). Stringent differential binding analysis (*n* = 3, adjusted *p*-value < 0.05) of the histone modification status of H3K27me3 and H3K4me3 showed significant differential binding, specifically proximal genes associated with AR signaling and NEPC signatures. These results indicated that *H19* overexpression in V16D significantly reduced the H3K4me3 differential binding on AR signature genes (e.g., *AR*, *NKX3.1*, *KLK2*, *ABCC4*, and *ZBTB10*) and altered the H3K4me3 differential binding on NEPC genes (e.g., *BRN2/POU3F2*, *KCNB2*, *BRINP1*, *SOGA3*, *CDH2*, and *REST*) (Fig. 7D, E, Supplementary Data 11). H3K27me3 binding was reduced on NEPC signature genes (e.g., *FGF9*, *HOXD10*, *RUNX1T1*, *MYT1*, and *ONECUT1*) (Fig. 7D, E, Supplementary Data 12). A small number of genes demonstrated significant changes in both histone marks (e.g., *BRN2/POU3F2*, *RGS7*, and *ETV5*). The observed histone changes in NE and AR signaling genes are concordant with previously described expression patterns in NEPC[35,45,46]. Bivalent regions are defined as regions of overlap between H3K4me3 and H3K27me3 marks for each cell line, and bivalently marked genes

are poised to regulate differentiation[47]. Bivalent genes enriched in histone modifications for V16D/H19 are shown in Supplementary Data 14. GO analysis of these genes revealed neuronal regulatory pathways among the top enriched pathways (Fig. 7F), indicating that *H19* overexpression provided a shift via chromatin remodeling towards NE differentiation.

**H19 knockdown induces alteration in genome-wide DNA methylation.** Epigenetic modifications have been shown to play a critical role in the progression to NEPC[45]. Since targeting DNA methylation is linked to histone methylation[48], we investigated the effect of *H19* knockdown on genome-wide methylation by enhanced reduced representation bisulfite sequencing to detect quantitative base-pair changes. Differential methylation analysis was done on OWCM-155 stably expressing shH19 or a control vector. This analysis demonstrated methylation changes in 8540 genes (59,982 sites; adjusted *p*-value < 0.01) with 3061 genes (12,766 sites) hyper-methylated and 5479 genes (47,216 sites) hypo-methylated (Fig. 8A, Supplementary Data 15). In total, 52% of the promoters were hypomethylated versus 27% hypermethylated (Fig. 8B), indicating the role of *H19* in driving gene expression through changes in promoter methylation.

To examine the clinical relevance, we compared the differentially methylated gene set (shH19 vs. control) with a previously established methylation gene set, which compared clinical samples of NEPC vs. AdPC[45] (Supplementary Data 16). After *H19* knockdown, the 541 genes with hypermethylation and 260 genes with hypomethylation were found to have reversed methylation status (Fig. 8C) from the ones established by Beltran

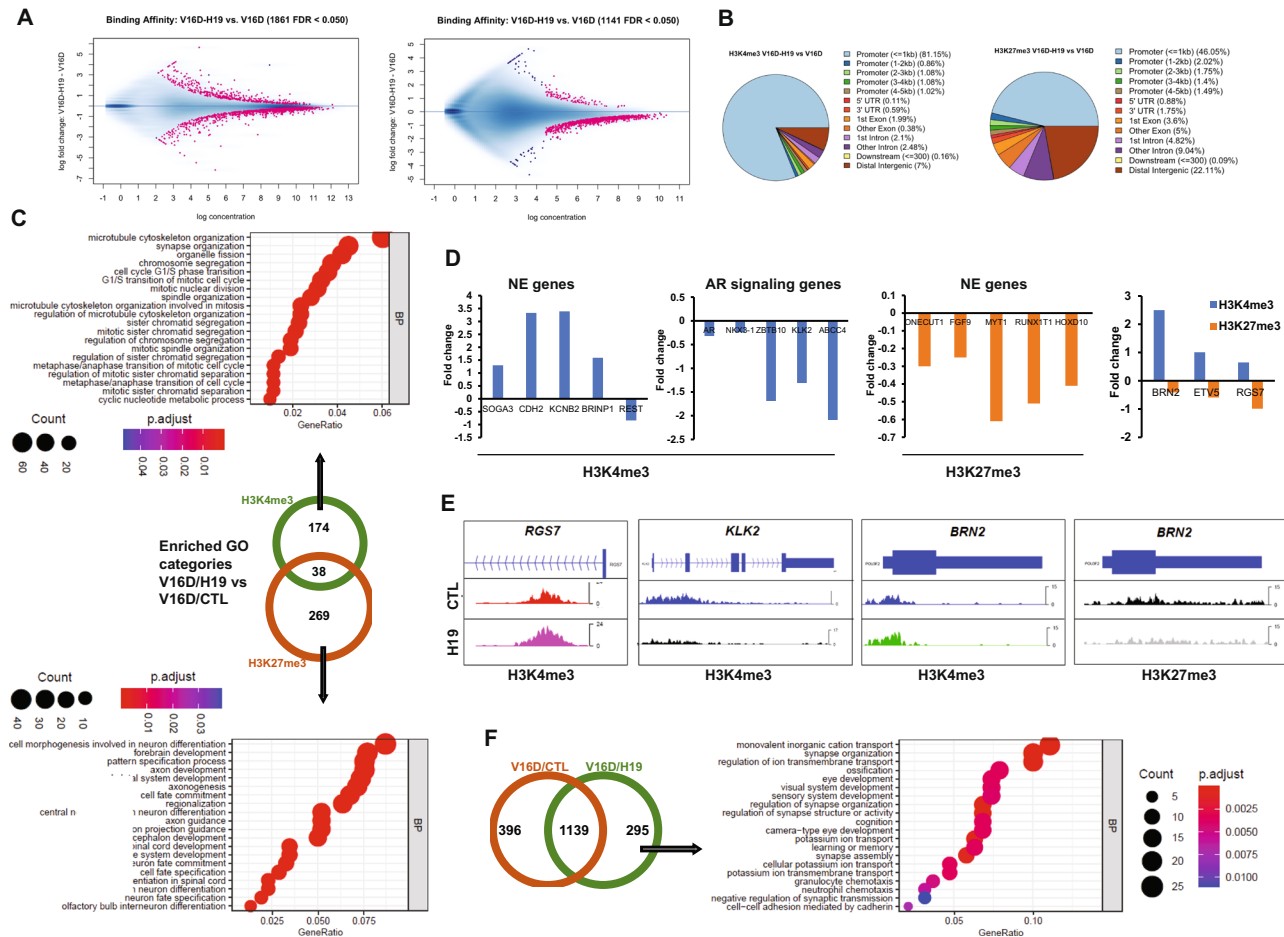

**Fig. 7 H19 reprograms the chromatin landscape of NEPC and AR signaling genes by altering the histone marks for H3K27me3 and H3K4me3. A** Binding affinity for significantly differentially bound sites (the first number in the parenthesis) in V16D/H19 vs. V16D/CTL cells. (Left: H3K4me3 mark, and right: H3K27me3 mark). **B** Distribution of proximal genomic features for significantly differentially bound sites in V16D/H19 vs. V16D/CTL cells for H3K4me3 and H3K27me3 histone marks. **C** Venn diagram (Center) of enriched GO categories from the differential binding results of V16D/H19 vs. V16D/CTL comparison for H3K4me3 and H3K27me3 histone marks. Arrows lead to dot plots with top 20 significantly enriched GO categories (pAdjustMethod = "BH", p-value Cutoff = 0.05, q valueCutoff = 0.05) for Biological Processes (BP) for H3K27me3 (bottom) and H3K4me3 (top). **D** Plot of representative genes in NE and AR signaling categories have significant differential binding (fold change) in V16D/H19 vs. V16D/CTL for H3K27me3 and H3K4me3 marks in regions proximal to or within their gene bodies. **E** Representative examples of ChIP-seq tracks for indicated genes and histone marks in V16D/H19 (H19) vs. V16D/CTL (CTL). **F** Venn diagram showing the overlap of bivalent sites from V16D/H19 and V16D/CTL. The right panel shows GO analysis (BP category) of the 295 bivalent sites enriched specifically for V16D/H19 only. In **C** and **F**, dot size indicates count. **A**–**F** Differential binding analysis data are collected from n = 3 independent biological replicates.

et al. This reversal in methylation status (Adj. p-value < 0.001) was found for genes associated with an NEPC signature[45] including *RGS7, CCND1, SPDEF, GATA2*; AR signaling genes, *TMPRSS2, PMEPA1*; NE phenotype genes, *CHGA*, and cell fate commitment genes, *ASCL1, HES5, KLF4, POU4F1* (Supplementary Data 16). Functional association analysis with GO analysis demonstrated that genes with hypermethylation are enriched for cell migration and neuron generation (p-value < 0.001, e.g., *HMX1, ENPEP, SOC2*, and *P2RY6*). Conversely, the GO pattern of those genes with hypomethylation fit into pattern specification processes, sequence-specific DNA binding, and negative regulation of cell differentiation (Fig. 8C, e.g., *RGS7, SPDEF, DLL3* and *NKX2.1*). The reversal of the methylation of these genes was present in the chromosomal loci observed from Beltran et al.[45] (Fig. 8D). Using the RNA extracted from organoids analyzed for methylation changes, gene expression was found to be decreased for hypermethylated genes, e.g., *P2RY6, SOCS3*, or increased for hypomethylated genes, e.g., *RGS7, ERG, CCND2, CDH4* (Fig. 8E). These results strongly point to the role of *H19* in regulating the methylation of genes associated with the NEPC phenotype.

**H19 is a putative diagnostic and predictive biomarker for NEPC.** Currently, there is an unmet clinical need for reliable diagnostic, prognostic, and predictive biomarkers for NEPC[45,49,50]. To test whether *H19* or other recently identified genes associated with NEPC were diagnostically useful, we analyzed sequenced clinical NEPC samples (n = 50, Table 1). The AdPC samples from this study were used as a control. For each NEPC sample and test gene, sensitivity and specificity analysis were performed. A gene/lncRNA was considered expressed or not expressed if it was >1 or <1 standard deviation from the control groups' expression, respectively. We tested genes related to AR-activity (*AR, KLK2*, and *PCA3*), commonly used NEPC markers (*CHGA*, SYP, and *NSE*), and four candidate test genes (*BRN2, SRRM4, PEG10*, and *H19*) (Fig. 9A). As expected, many samples showed no AR expression or activity. Concerning NEPC markers, *SYP* had the greatest sensitivity (90%), and *NSE* had the greatest specificity (87%). Notably, the four test transcripts, *BRN2, SRRM4, PEG10*, and *H19*, had sensitivities of 86%, 88%, 92%, and 86%, respectively, and performed similarly to NEPC markers. However, with their relatively lower specificities, they would best

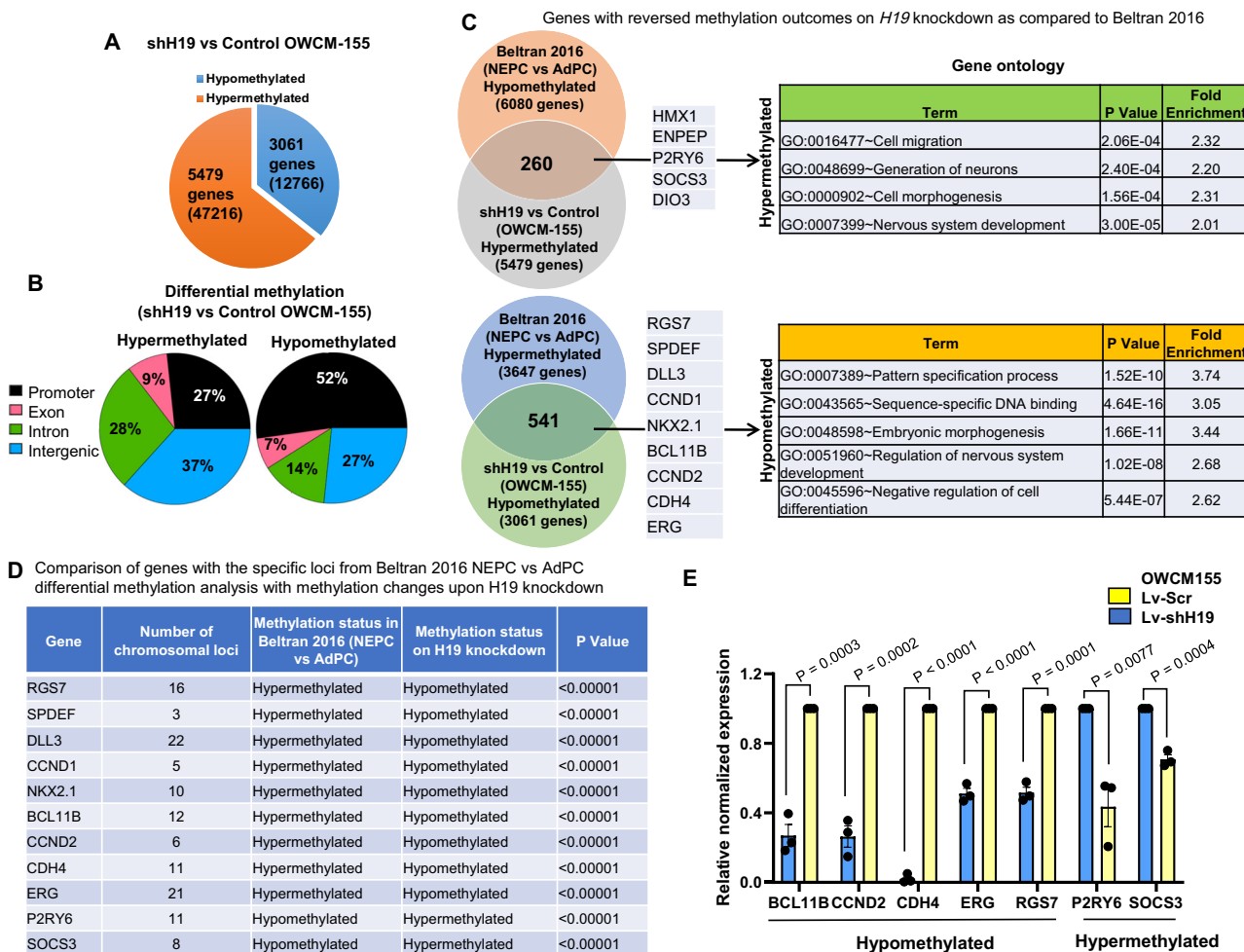

**Fig. 8 Genome-wide methylation analysis of H19 knockdown in NEPC organoid. A** Pie chart showing the number of differentially methylated genes (shH19 vs. control OWCM-155, n = 3 biologically independent replicates), identified by annotating hyper- and hypomethylated loci (the number is reported in parentheses) (p-adj. < 0.01). **B** Pie chart depicting the percent DNA methylation-based on indicated gene regions. **C** Left, Venn diagram of comparison of shH19 vs. control OWCM-155 differential methylation gene set vs. the Beltran 2016 differential methylation gene set, NEPC vs. AdPC. The common area depicts the number of genes with reversal of methylation outcomes. Middle, example lists selected genes with reversed methylation upon *H19* knockdown. Right, selection of functional categories enriched after analysis of differentially methylated genes (*p*-value < 0.001). **D** Number of chromosomal loci of selected genes with reversed methylation outcomes upon *H19* knockdown (*p* value < 0.005). See Methods for statistical determinations (**A**–**D**). **E** Gene expression analysis of differentially methylated genes in OWCM-155 NEPC organoid with shH19 (Lv-shH19) compared to control (Lv-Scr). Data are mean ± SEM (**E**); n = 3 biological replicates. *p* Values were calculated by unpaired two-tailed Student *t* test (**E**).

serve in a panel with more established NEPC markers. For example, in patients where *CHGA* was negative (Fig. 9A—black/red arrows) or both *CHGA* and *SYP* were negative (Fig. 9A—red arrows), *H19* positively detected NEPC. This data supports incorporating these "next-generation NEPC markers", including *H19*, to be used clinically to enhance the ability to detect NEPC.

It is estimated that 20–30% of metastatic CRPC tumors develop tNEPC[51]. We explored whether *H19* might predict the clinical outcome using the Decipher GRID database, focusing on AdPC samples from the MCII cohort (n = 232, Table 1). MCII represents tumors primarily with unfavorable pathology (i.e., high grade/stage) and long-term follow-up for treatment and outcomes (median 18 years). This cohort contains samples treated with radiotherapy (RT), adjuvant ADT, or post-radical prostatectomy[34,52]. We performed survival analysis to establish *H19*'s ability to stratify patients with an increased probability of biochemical recurrence (BCR) or metastasis (MET) as their clinical end-point. ADT-treated patients were grouped by *H19* expression into tertiles. We observed that samples with the highest tertile of *H19* expression vs. mid or low levels had a

significantly higher probability of BCR or MET (Fig. 9C—*p*-value = 0.00996 and *p*-value = 0.0162, respectively). With *H19* expression stratified in the same manner, we also generated Kaplan–Meier curves in untreated patients. Unlike ADT-treated patients, untreated patients did not significantly differ in the probability for BCR nor MET (Fig. 9B—*p*-value = 0.627 and *p*-value = 0.880, respectively). Taken together, in patients that receive ADT and consequently have a higher probability of tNEPC, an elevated *H19* level is a predictive biomarker for poor survival-related outcomes.

## Discussion

We discover the lncRNA *H19* as a driver of PCa lineage plasticity and induction of the NE phenotype. 122 lncRNAs distinguish NEPC from AdPC, and *H19* is one of the most highly expressed lncRNAs within this signature[28]. Importantly, elevated levels of *H19* are shown to be associated with higher Gleason grade and neoadjuvant hormone therapy, suggesting its role in PCa progression. These findings are experimentally validated in pre-

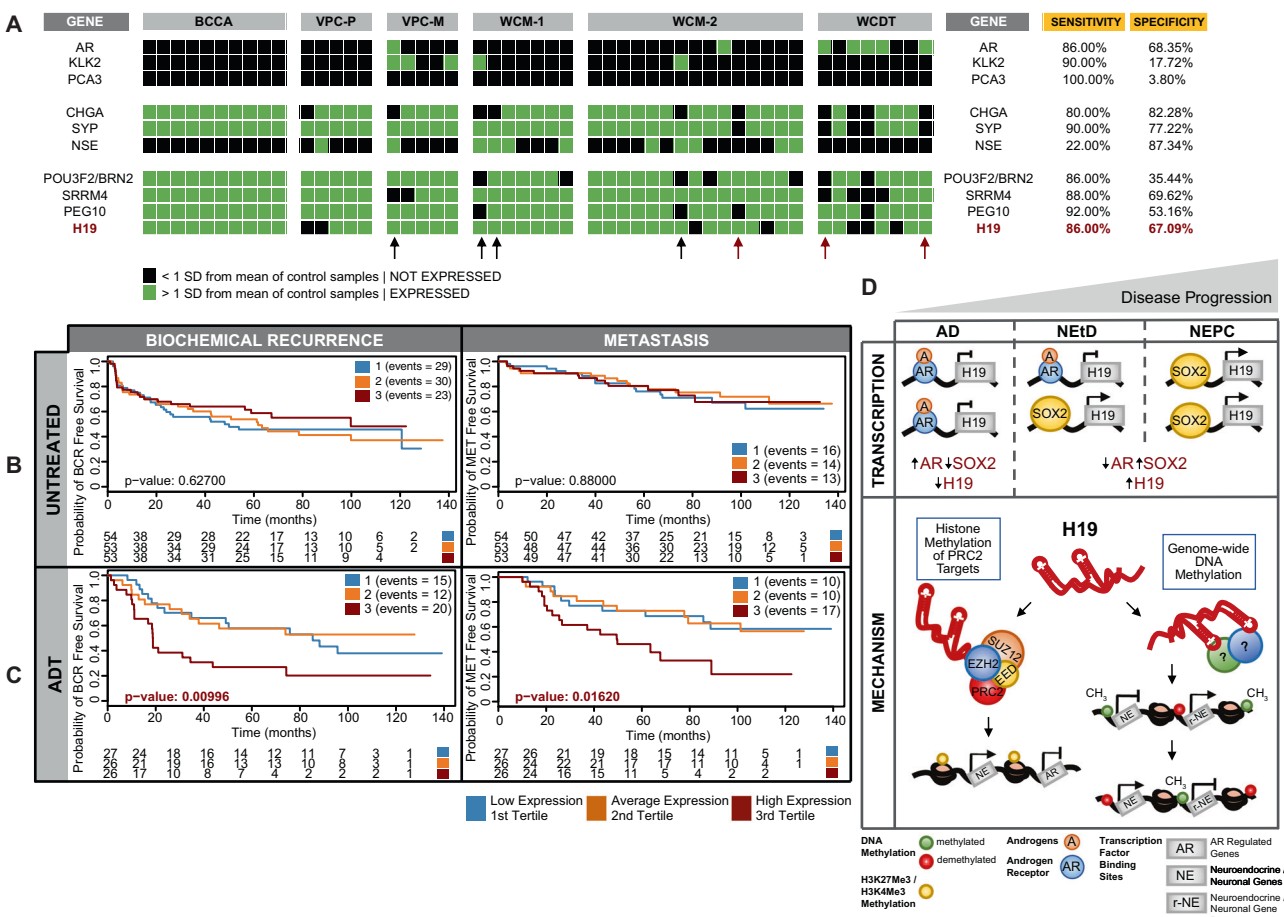

**Fig. 9 Diagnostic and predictive ability of H19. A** Assessment of sensitivity and specificity of *H19* in NEPC for use as a diagnostic along with other recently identified NEPC oncogenes (BRN2, SRRM4, and PEG10). Arrows denote patients that are negative for CHGA (black) or CHGA, SYP, and NSE (red) yet positive for *H19*. **B**, **C**, Kaplan–Meir estimates for the impact of *H19* on prostate cancer clinical outcomes. **B** Survival analysis for *H19* in ADT untreated patients and **C** ADT treated patients for biochemical recurrence (BCR—left panels) and metastasis (MET—right panels). Samples were split by *H19* tertile expression to test if either of these subgroups of patients were more likely to develop BCR and MET in the context of treatment. P values were calculated using a Kaplan–Meier estimator statistical test to determine significance for the observed stratification in probabilities across the three subgroups in each panel. **D**, Illustration of *H19* transcription and mechanism of action during PCa, NEtD, and tNEPC. During early stage disease, adenocarcinoma cancer cells are driven by an active AR bound to androgens that prevent the transcription of *H19*. As the disease progresses (gray triangle), ADTs suppress AR activity, *SOX2* increases, and results in the persistent transcription, expression, and activation of *H19*. Once active, *H19* operates via dual epigenetic mechanisms; (on the left) binds PRC2 complex members, aiding in altering methylation on H3K4Me3/H3K27Me3 histone marks of PRC2 target genes and (on the right) genome-wide alteration of methylation at CpG sites on DNA. Collectively, these epigenetic changes result in the activation of NE gene expression and deactivation of AR signaling gene expression.

clinical models of NEPC, including patient-derived NEPC organoids, and murine *Trp53/Rb1*, *Trp53/Pten*, and *Rb/Pten* DKO organoids. In addition, our sequence analysis identifies the longest isoform of *H19* as the active variant in NEPC and highly conserved, both in sequence and secondary structure across multiple species.

NEtD resulting in a transition of CRPC to NEPC occurs through an intermediary stem-like state in which cells exhibit EMT and stem cell-like features[39]. This lineage plasticity is thought to be reversible. Our data confirmed the potential for *H19* to be a central regulator of this bidirectional phenotype creating a stem cell-like permissive environment for lineage plasticity. This is shown by *H19* leading to concomitant changes in NE gene expression, while the reduction of *H19* expression in *Trp53/Rb1* DKO organoids induced a lineage reversal from a NE-like to a luminal-like phenotype. This suggests that *H19* aids cells in acquiring a stem cell-like state that may be essential for lineage plasticity and NEtD. Extensive study of miR-675, a mediator for *H19* function and hosted within *H19*, has been previously carried

out[53]. However, our results in murine prostate organoids indicated that elevated miR-675 does not alter the expression of NE genes (Supplementary Fig. 14B–D), suggesting that *H19* does not function via miR-675 in our system. The observation that persistent nuclear AR expression and median serum PSA levels (>60 ng/ml) occur in a subset of small cell NEPC patients suggests that the AR is only part of the control mechanism regulating NEPC function[3]. Lineage reversal mediated by manipulating *H19* and EZH2 to restore AR signaling might result in a therapeutically targetable phenotype derived from this aggressive disease.

The control of *H19* transcription appears to be complex. The role of androgens in controlling *H19* has previously been reported[54], and we corroborated that *H19* transcription is suppressed by androgen. During ADT, SOX2 levels rise and bind to the *H19* promoter, increasing *H19* transcription, potentiating the induction of the NE phenotype. The combined loss of *TP53* and *Rb1* inducing *H19* expression was consistent with the observation that TP53 and RB1 control the level of SOX2 in PCa cells[16]. Since

our data showed that knockdown of SOX2 or OCT4 decreased *H19* levels, a feed-forward mechanism may exist in PCa, in which increases to stem cell factors enhance the transcription of *H19*, and then *H19* drives further elevation in these stem cell genes. The promoter region of *H19* contains multiple putative TF binding sites, including TP53, E2F, and HIF1α[55–58], which could also regulate this lncRNA. Recently, we have shown in multiple cancers that the Pim kinases regulate *H19* levels suggesting Pim could play a role in NEtD[59].

*H19* is a maternally imprinted gene with its expression closely linked to *IGF2* through regulation mediated by the ICR1 locus. Studies have shown the loss of *H19/IGF2* imprinting in cancer[60,61]. Our methylation assay did not find a loss of imprinting in NEPC (Fig. 3C), and increased IGF2 expression in NEPC was detected compared to *H19* (Supplementary Figs. 11F, 14A, and 15A–D). Positive correlation patterns within our clinical samples (Supplementary Fig. 15E) suggest these genes are co-regulated. However, in bladder cancer, SOX2 stimulates IGF2 expression[62], and with SOX2 elevation occurring due to androgen withdrawal, this may further elevate the transcription of *IGF2* in NEPC.

Because EZH2 plays a prominent role in NEPC regulation[63–65], we examined whether *H19* may control the levels of a diverse set of genes by interacting with the PRC2 complex. Our data demonstrated that *H19* overexpression in multiple PCa cell types increased H3K27me3 and H3K4me3 levels. In addition, we showed that *H19* could bind PRC2 complex members, suggesting a partial mechanism for this effect. LncRNAs, e.g., HOTAIR, can bind to the PRC2 complex and interact with LSD1, a demethylase for H3K4me2, leading to gene activation[66]. However, in preliminary experiments, we have not seen direct binding of LSD1 by *H19*. Similar to HOTAIR, *H19* may also function as a modular bifunctional RNA, but further experiments are required to identify other *H19* binding partners. ChIP-seq data further demonstrated a role for *H19* as an epigenetic modifier. Histone marks were switched from a transcriptionally repressive state to an active state for NEPC signature genes and for androgen signaling genes histone marks went from active to a repressive state. Our analysis identified several bivalent genes with a potential role in the NEPC transition poised for further regulation. For example, KDM5A, which directly interacts with the PRC2 complex in embryonic stem cells to promote a transcriptionally repressive state during differentiation[67]. It is possible that *H19* acts as a scaffold for KDM5A and EZH2. Thus, further studies are warranted to investigate the exact mechanism of chromatin reprogramming by *H19*.

Genome-wide single cytosine DNA methylation analysis has previously shown strong epigenetic segregation between NEPC and AdPC subtypes within patient samples[45]. Compared to AdPC, DNMT1, 3B, and 3A are elevated in NEPC[45] and could drive this change. In addition, our data demonstrate that *H19* knockdown induced significant differences in the DNA methylome, both hypo- and hypermethylation, with many identical chromosomal loci targeted that had been previously identified in a comparison of NEPC with AdPC[45]. These findings pointed to a complex mechanism by which *H19* regulates histone modification and DNA methylation, two hallmarks of epigenetic regulation.

Survival analysis of patient cohorts showed that patients treated with ADT increased the probability of developing biochemical recurrance or metastasis when *H19* was elevated. In ADT untreated patients, *H19* levels did not show a difference in probability for biochemical recurrance nor metastasis-free survival. This result is consistent with our in vitro observations that tNEPC induced by androgen blockade is associated with higher *H19* levels. In addition, our results demonstrate that *H19* performed comparably to other NEPC biomarkers used for immunohistochemistry-based diagnosis. Given the rapid advances in blood-based liquid biopsies, recently shown efficacy in NEPC[68], and the relative stability of lncRNAs, *H19* levels could be an essential contributor to the diagnosis of tNEPC in ADT treated patients.

In summary, we show that *H19* is highly expressed in patients with NEPC, a putative diagnostic and predictive marker associated with disease outcome, and a regulator of NE and AR signaling associated with the induction of NEPC. Most significantly, we show evidence that it drives lineage plasticity from a luminal to NE phenotype, which upon H19 knockdown reverses this transition and results in tumors becoming ADT sensitive. For these reasons, this lncRNA warrants serious consideration in the clinical management of patients on ADT at risk of tNEPC, and as a therapeutic target for reversing tumor plasticity, which induces a treatable form of PCa.

## Methods

**Clinical patient samples and cohorts.** We used nine clinical cohorts, five sequenced and four profiled by microarray (Table 1). Sequenced cohorts were from (1) Vancouver Prostate Center patients (VPC-P) and model systems (VPC-M); (2) Weill Cornell Medicine (WCM-1 and WCM-2); (3) West Coast Dream Team (WCDT); and (4) British Columbia Cancer Agency (BCCA). Microarray cohorts were from the Mayo Clinic (MCI and MCII), Johns Hopkins School of Medicine (JHMI), and samples from the Decipher Genomic Resource Information Database (GRID) housed at Decipher Bioscience Inc. Sequenced cohorts totaled 230, microarrayed totaled 27,695, and cumulatively our study utilized 27,321 samples. For the VPC, 84 specimens were collected as previously described[21,28,69,70] and amalgamated for this study. For WCM-1 and WMC-2, 37, and 49 specimens were collected as previously described (WCM-1[45] and WCM-2[18]), respectively. For WCDT, 45 specimens from a larger cohort of 200–300 (collection ongoing were collected as previously described[3,71]. For BCCA, 15 specimens were collected as previously described[72]. Cohorts MCI and MCII, a total of 813 samples were collected as previously described (MCI[73] and MCII[52]). JHMI samples, totaling 33 samples were retrieved from surgical pathology and consultation files of Johns Hopkins Hospital (John Hopkins Registry) from 1999 to 2013, as previously described[74]. The 33 samples were annotated originally as six morphologically diagnosed pure prostate small cell carcinoma samples (SCPC), 12 high risks (Gleason 9–10) Adenocarcinoma (AdPC), 10 SCPC (SC-mixed), and 5 AdPC (AdPC-mixed) from mixed histology tumors containing separate AdPC and small cell components. For this cohort, samples were re-classified by their genomic signature as previously described, 10 SC/NE-like (NEPC), 10 Mixed Pathology (MX-P), and 13 Adenocarcinoma (AD). GRID prospective samples, a total of 26,245 (16,806 from radical prostatectomy (GRID-RP) and 9439 from biopsy tissue (GRID-BX) were collected from the clinical use of the Decipher test and previously described[35].

**Cell culture.** HEK293T, LNCaP, C4-2B, VCAP, PC3, LASCPC-01[75], and NCI-H660 cell lines are from the American Type Culture Collection (ATCC). The cell lines were cultured as recommended by the ATCC. Dr. Amina Zoubeidi (University of British Columbia, Vancouver, BC) provided V16D$^{CRPC}$, 42D$^{ENZR}$, and 42F$^{ENZR}$ cells, which were cultured as described previously[18]. LAPC-4 cells (RRID: CVCL_4744) were provided by Dr. Charles Sawyers (Sloan Kettering Memorial Center, NY, USA) and were cultured as described previously[39]. Regular testing of mycoplasma contamination was performed in these cell lines with MycoAlert™ Mycoplasma Detection Kit (LT07-118, Lonza), and only mycoplasma-free cells were used for experimentation.

**Organoid culture.** Dr. Himisha Beltran provided NEPC patient-derived organoids- OWCM-154, OWCM-155, OWCM-1078, and OWCM-1262 and cultured as described previously[37]. Prostate cancer biopsies were provided by the University of Arizona Cancer Center Tissues Acquisition and Cellular/Molecular Analysis Shared Resources, and the study was conducted under the University of Arizona Institutional Review Board approval as previously described[76]. Since the patient-derived samples were de-identified, the human research review determined the study as not human subject research. The cultures were replenished with fresh media every 3–4 days during organoid growth. Dense cultures with organoids ranging in size from 200–500 µm were passaged weekly. For murine organoids, all animal experiments were performed in accordance with protocols approved by The University of Arizona Institutional Animal Use and Care Committee (IACUC). Following established procedures[77], cells were collected from mouse prostates and cultured in growth factor-reduced Matrigel in ADMEM/F12 along with EGF, Noggin, and R-spondin (ENR) supplements. The organoids are collected and trypsinized, and smaller cell fractions are then incubated in a plate containing ENR-supplemented media. Prostate organoid cultures were bio-banked using Bambanker (Gibco) at −80 °C.

**Patient-derived organoid xenograft studies**. Totally, 500,000 cells derived from OWCM-155 NEPC organoids (shSCR and shH19 groups, $n = 4$ mice per group) were injected with Matrigel (Corning) 1:1 subcutaneously into NOD SCID gamma (NOD.Cg-Prkdc$^{scid}$ Il2rg$^{tm1Wjl}$/SzJ) male mice (Jackson Laboratories, Bar Harbor, Maine). Mice used for xenografts were 5–7 weeks old. The mice were housed in ventilator racks (RAIR IVC system, Lab Products Seaford, DE) and maintained under specific pathogen-free conditions. The mice were fed NIH-31 irradiated pellets (Tekland Premier, Madison, Wisconsin), and sterile water was freely available. Daily light cycles were kept consistent in the animal facility (12 h light and 12 h dark). Cages were changed entirely once a week. Sentinel mice were screened monthly by ELISA serology for mycoplasma, mouse hepatitis virus, pinworms, and Sendai virus and tested negative. Tumor volume was measured every week with a caliper. After 2–3 months of tumor growth, when the largest tumor size reached the maximum allowable tumor burden, the mice were euthanized in a $CO_2$ chamber. The tumors were then excised, collected, and weighed. Part of the tumor harvested was fixed in 10% neutral buffered formalin, paraffin-embedded, and subjected to immunohistochemical staining for various markers. The other part of the harvested tumor was used for RNA extraction. Animal care and experiments were carried out in accordance with the University of Arizona IACUC guidelines.

**Lentiviral plasmids**. Knockdown of human *H19* was performed using the lentiviral plasmids pLenti-siH19-GFP (Abcam, #i009382) and pLenti-scrambled siRNA-GFP (Abcam, #LV015-G) was used as a control. These siH19 plasmids allowed for direct non-viral plasmid transfection for immediate expression (siH19) and packaged into lentiviral particles for high-efficiency transduction and stably integrated expression (shH19). Of the four siRNA target sequences we tested, two (shH19-C and shH19-D) demonstrated a functional knockdown.

  shH19-A 1483: GAAGCGGGTCTGTTTCTTTACTTCCTCCA
  shH19-B 1551: ACCCACAACATGAAAGAAATGGTGCTACC
  shH19-C 1589: CCTGGGCCTTTGAATCCGGACACAAAACC
  shH19-D 1710: CCTCATCAGCCCAACATCAAAGACACCAT

For all experiments, shH19-C was used unless indicated otherwise. Overexpression of human *H19* was performed using pLenti-GIII-CMV-H19-GFP-2A-Puro (Abm, # LV178008). For *H19* knockdown in mouse organoids, plasmid GIPZ Mouse *H19* shRNA purchased from Dharmacon (RMM4431), and for Cre recombinase expression, the plasmid FUGW-Cre (Addgene) was kindly provided by Dr. Owen Witte (UCLA, Los Angeles, CA). shOCT4 (LL-hOCT4i-1) (Addgene plasmid # 12198; http://n2t.net/addgene:12198; RRID:Addgene_12198) and shNANOG (LL–hNANOGi) were a gift of George Daley (Addgene plasmid # 12196; http://n2t.net/addgene:12196; RRID:Addgene_12196)[78]. shSOX2 (pLKO.1 Sox2 HM) was a gift from Matthew Meyerson (Addgene plasmid # 26353; http://n2t.net/addgene:26353; RRID:Addgene_26353)[79]. shP53 (pLKO-p53-shRNA-941) (Addgene plasmid # 25637; http://n2t.net/addgene:25637; RRID:Addgene_25637)[80] and shRB1 (pLKO-RB1-shRNA19) (Addgene plasmid# 25640; http://n2t.net/addgene:25640; RRID:Addgene_25640)[81], were a gift from Todd Waldman.

**Reagents**. 5α-Dihydrotestosterone (DHT) (Cat. no. D-073-1ML) was purchased from Sigma. Enzalutamide (MDV3100) (Cat. no. S1250) was purchased from Selleckchem. Doxycycline hydrochloride (Cat. No. BP-2651) was purchased from Sigma.

**Primers**. Primers for qPCR and ChIP qPCR are listed in Supplementary Data 17.

**Cell viability (XTT) assay**. LNCaP with stably expressing control and H19 knockdown conditions were seeded into 96-well plates at a density of 5000 cells per well and were allowed to grow for 72 h. Following the manufacturer's protocol, cell viability was measured using XTT cell proliferation assay (Trevigen Cat # 4891-025-K). For testing Enzalutamide (ENZA) effect on cell proliferation, LnCaP cells with H19 overexpression or H19 knockdown with or without TP53/Rb1 deletion were seeded at 5000 cells/well. ENZA was added at indicated doses for 72 h, using DMSO as control. Following the manufacturer's protocol, cell viability was measured using XTT cell proliferation assay (Trevigen Cat # 4891-025-K).

**Lentiviral particle production**. Lentiviral particle production and infection were performed as described previously[82]. Briefly, lentiviral vectors were co-transfected with psPAX2 and pVSVG vectors into HEK293T cells. Supernatants were collected at 24 and 48 h after transfection, concentrated by ultracentrifugation in SW28 (Beckman) rotor and stored at −80 °C. For infection of adherent cells, $10^6$ cells per well were seeded in six-well plates and infected with concentrated lentiviral particles 24 h post-seeding. Established procedures were used for lentiviral transduction of organoids by spinoculation[37]. For viral transduction protocol, control lentivirus was used at the same titer as the experimental virus ($10^8$ TU/ml). The transduced organoids were passaged at least twice before the cell growth assays were performed to minimize lentiviral toxicity effects.

**Doxycycline inducible system for H19 expression**. For inducible overexpression of H19 (full length, H19$^{FL}$, and EZH2 binding site deletion mutant fragment, H19$^{DEL}$), plasmid Tripz-H19$^{FL}$ and Tripz-H19$^{DEL}$ was constructed. This was done by PCR of pLenti-H19 plasmid for full length and the shorter segment of H19 to generate H19$^{FL}$ and H19$^{DEL}$. These fragments were then individually cloned into the Tripz vector to generate the inducible expression vector. The Tripz-H19$^{FL}$ and Tripz-H19$^{DEL}$ were then transfected in HEK293T cells to generate lentivirus following the above method. The C4-2B or LNCaP cells were then transduced with these lentivirus vectors and selected with puromycin (1 μg/ml) to generate stable cell lines.

**miR-675 expression system**. To express H19 fragments in mouse organoids, we used a retroviral vector expressing full length and Middle H19 fragment (741–1407) encoding miR-675 of H19[83] (kind gift from Dr. Anindya Dutta). pMSCV retro was used as a control. Retrovirus encoding these fragments was produced, and murine prostate organoids were transduced with the control and H19 fragments. After stable transduction of these organoids post-two passages, we extracted the RNA, and qPCR was performed.

**Real-time PCR and gene expression analysis**. Total RNA was isolated from cells using TRIzol reagent (Invitrogen, 15596-018). For organoids, 700 μl RLT buffer from an RNeasy mini kit (Qiagen) was added to each well and was gently titrated to dissolve the matrigel and then collected in microfuge tubes for further RNA extraction following the manufacturer's protocol. One microgram of total RNA was reverse transcribed using an i-Script cDNA Synthesis System kit (Biorad, 1708891) following the manufacturer's protocol. qPCR primer pairs were selected, and at least three primer pairs per gene were tested before using them for experiments. Primer sequences are shown in Supplementary Data 17 and original Ct values of H19 are shown in Supplementary Data 18. To measure gene expression, real-time PCR was performed using SsoAdvanced™ Universal SYBR® Green Supermix (Biorad, 1725271), following the manufacturer's protocol. Expression levels of each transcript were quantified by using Bio-Rad CFX96 Real-Time PCR Detection System. 18S values were used as endogenous controls for human samples, and Actin and Hprt were used for mouse samples. For Taqman based assays for miR-675, RNA was extracted as above. One microgram of total RNA was reverse transcribed using Taqman MicroRNA reverse transcription kit (Life Technologies, Invitrogen) following the manufacturer's protocol. Primers for murine miR-675-6p, miR-675-3p, and endogenous control U6 snRNA are listed in Supplementary Data 17. qPCR was performed using Taqman Fast advanced master mix (Life Technologies, Invitrogen) following the manufacturer's protocol.

**Immunohistochemistry (IHC) staining**. IHC for CK8 (Anti-Cytokeratin 8 antibody [EP1628Y] - Cytoskeleton Marker (ab53280), 1:200 dilution) and Ki67 (Ki-67 (D2H10) (IHC Specific) Catalog no. 9027, 1:200 dilution) was performed using Leica Bond RXM system (Leica). Formalin-fixed mouse organoid samples were deparaffinized with xylene/ethanol, and then antigen retrieval was performed using Citrate Retrieval Buffer (AR9961, Leica), heat-induced at pH 6.0 at 98 °C for 20 mins. After washing, slides were incubated with primary antibody for 15 min at RT. Antibodies used were Anti-Cytokeratin 8 antibody [EP1628Y] - Cytoskeleton Marker (Catalog no. ab53280), 1:150 dilution and Ki-67 (D2H10) Rabbit mAb (IHC Specific) Catalog no. 9027, 1:200 dilution. Post-primary antibody was applied as needed (HRP conjugated IgG, DS9800, Leica) for 8 min. A post polymer signal was amplified. DAB staining kit (DS9800, Leica) was used for final detection, counterstained with hematoxylin (5 min), and mounted in a non-aqueous solution (Leica Micromount, 3801730). The images were captured using a binocular Leica light microscope (Leica™ DM2500) at the bright field and a CCD color video camera (Leica DFC320) attached to a computer system.

**Subcellular fractionation of RNA**. Followed NE-PER™ Nuclear and Cytoplasmic Extraction kit protocol (Thermo cat# 78833) and added 1:10 RNaseOUT™ Recombinant Ribonuclease Inhibitor (Thermo cat# 10777019) to the nuclear and cytoplasmic fractions. Extracted RNA using RNAzol protocol (Sigma cat# R4533).

**Chromatin immunoprecipitation (ChIP)**. ChIP assay was performed using the SimpleChIP® Plus Enzymatic Chromatin IP Kit (Cell Signaling cat # 9005) according to the manufacturer's protocol. Chromatin was fragmented using the Bioruptor® Pico sonication device (Diagenode). Equal volumes of chromatin were immunoprecipitated with either antibody against AR (Cell Signaling, cat # 5153), SOX2 (Cell Signaling cat # 23064S; SantaCruz Biotech cat # sc-365823), or mouse or rabbit IgG as a negative control. Primers for each binding site were listed in Supplementary Data 17.

**Luciferase assay**. pGL4-H19 (minimal promoter 0.8 kb) reporter plasmid (H19-PP) was generously provided by Dr. Karl Pfeifer (NIH/NICHD) and Dr. Jie Chen (University of Illinois at Urbana-Champaign). Sox2 binding site mutant H19 was generated using the oligonucleotide primers, sense 5′-CACAGGGGACTCCC CTCTGTCACCAGACCCTCCCTCTTCAG-3′ and antisense 3′-CTGAAG AGGG AGGGTCTGGTGACAGAGGGGAGTCCCCTGTG-5′, and QuikChange®

Lightning Site-Directed Mutagenesis Kit according to the manufacturer's protocol. Cells were plated in 24-well plates and transfected with PGL4-luc containing wild-type or mutant H19 promoters using Lipofectamine 3000 (Invitrogen), and pRL-TK was used as an internal control. Luciferase activity was measured by Dual-Luciferase Assay reagent (Promega) based on the manufacturer's manual.

**DNA methylation detection**. MethylMeter assays for molecular beacon-based detection were designed and performed as described previously[38]. DNA samples were cleaved with MseI and were fractionated without purification. Fragmented samples were separated into methylated and unmethylated fractionations with the MethylMagnet® kit (cat# MM101K, RiboMed Biotechnologies) following the manufacturer's protocol. A target-specific primer with a 5′ truncated promoter extension (5′ CTTACAATGCATGCTATAATACCACTATCGGTGCTTTATTTA AGCGCGGAATTTGCTGTGCTCAT) and a reverse primer (5′ AGTGAATAAG GCTTGCCCTGACGAGGACTCAAGTCACGCCTA CC) targeted a 1624 nt MseI fragment located 365 nt upstream of the H19 long-variant 1 RNA TSS. CAPS detection reactions were performed as described earlier (1). Amplicons with a full-length promoter were made with a promoter-specific primer (5′ CCTTTAAAGA AAATTATTTTAA ATTTATGTTTGACAGATCTTACAAGTGCATGCTATAATA CCA) and a universal reverse primer (5′ AGTGAATAAGGCTTGCCCTGACGA) as previously described. The H19-specific annealing temperature was 60.7 °C. Fluorescence signals were generated when abortive transcripts from the synthetic promoters contributed to opening a molecular beacon. Methylation results were expressed as a percentage of the methylated DNA signal divided by the sum of the methylated and unmethylated signals. As a control sample, DNA extracted from the urine sample of a 42-year-old healthy male was used.

**Western Blotting**. Total protein was extracted from adherent cells grown in vitro and organoid cultures as described previously[76]. To isolate histone proteins from cells and organoid cultures with post-translational modifications intact, EpiQuik Total Histone Extraction Kit (Epigentek, OP-0006-100) was used according to the manufacturer's protocol. Antibodies used: SOX2 (sc-365823, Santa Cruz Biotechnology, 1:400 dilution), H3K27me3 (9733S, Cell Signaling Technology, 1:1000 dilution), EZH2 ((D2C9) XP Rabbit mAb (5246, Cell Signaling Technology, 1:1000 dilution), NSE (sc-21738, Santa Cruz Biotechnology, 1:500 dilution), Synaptophysin (sc-365488, Santa Cruz Biotechnology, 1:500 dilution), CHGA (60893S, Cell Signaling Technology, 1:1000 dilution), BRN2 ((D2C1L) Rabbit mAb 12137, Cell Signaling Technology, 1:1000 dilution), H3 ((D1H2) XP® Rabbit mAb 4499, Cell Signaling Technology, 1:1000 dilution), Androgen receptor (5153S, Cell Signaling Technology, 1:1000 dilution), H3K4me3 (ab8580, abcam, 1:1000 dilution), P53 (2527 S, Cell Signaling Technology, 1:1000 dilution), Rb (9313S, Cell Signaling Technology, 1:1000 dilution), CK8 (sc–57004, Santa Cruz Biotechnology, 1:300 dilution), PSA (sc-7316, Santa Cruz Biotechnology, 1:250 dilution), HRP conjugated anti-β-actin (Cat. no. A3854, Sigma, 1:10,000 dilution). HRP-linked mouse IgG (Cat. no. NA931V, GE Healthcare Life Sciences, 1:10,000 dilution) and HRP-linked rabbit IgG (Cat. no. NAV934V, GE Healthcare Life Sciences, 1:10,000 dilution). The levels of proteins were quantified by dividing the intensity levels with ACTIN levels and then normalizing the levels to their respective control samples using ImageJ software.

**Cell and organoid growth assay**. Assessment of proliferation was conducted using the IncuCyte system. Briefly, after 48-hour siRNA treatment, cells were passaged, counted, and seeded at 2,000 cells/well in replicate on a 96-Well Plate (Corning) on day 1, with IncuCyte readings taken at 24-h cycles starting from day 0. Media was replenished on day 7. Confluence area calculations made by the IncuCyte algorithm were normalized to day 0 and analyzed using GraphPad Prism Software.

To measure organoid growth, organoids were dissociated with TrypLE (Invitrogen) into tiny cell clusters, and 5000 cell clusters were plated per well and then incubated for six days using DMSO as control. A real-time imaging system (IncuCyte) was used to measure cell proliferation using the organoid module. The images were captured every 12 h. The percentage confluence of organoids was plotted against time for organoid viability analysis.

**Transwell invasion assays**. H19 knockdown mouse TP53/RB1 organoids and their corresponding control cells were placed in a Corning fluorblock 24-well Transwell plate (8-mm pore size; Corning) as previously described[84]. Briefly, organoids were dissociated into cell clusters, and these clusters (10⁴/well/condition) were suspended in 200 ml of matrigel/ADMEM (1:3) and added to the upper chamber of transwell inserts. The lower well was filled with 500 μl of mouse prostate organoid media in contact with the insert membrane. The cells were allowed to migrate for 72 h post-plating. Images of the bottom of the insert were captured for migrated cells, and relative GFP fluorescence was measured by ImageJ (NIH) at four different microscopic fields.

The invasive abilities of cell models were assessed by using Matrigel-coated 24-well plate inserts (Corning® BioCoat™ Matrigel® Invasion, Corning, NY) according to the manufacturer's instructions. Briefly, 20,000 cells were seeded in the top chamber of a Matrigel-coated 24-well plate inserts in a serum-free medium. Totally, 10% FBS was added to the lower chamber as a chemo-attractant. After

20 h, cells were fixed and stained with DAPI, the filter was fluorescently imaged, and the cells remaining on the filter counted using ImageJ software.

**RNA immunoprecipitation**. A RIP assay was carried out using a Magna RIP RNA Binding Protein Immunoprecipitation Kit (Millipore; Cat# 17-700) according to the manufacturer's instructions. Briefly, LASCPC-01, control vector or H19 over-expressing LNCaP and V16D cell lines were grown in 15 cm culture dishes, and approximately $20 \times 10^6$ cells were harvested with ice-cold PBS and pelleted to 5 min, 1500 rpm, 4 °C. The resulting pellets were lysed in RIP lysis buffer containing protease inhibitor cocktail and RNase inhibitor, followed by centrifugation at 14,000 rpm, 4 °C for 10 min. An aliquot of the resulting supernatant was incubated with magnetic beads pre-conjugated with EZH2 (Cell Signaling; Cat. No. 5246 S), SUZ12 (Cell Signaling; Cat. No. 3737 S) or rabbit IgG (Cell Signaling; Cat. No. 2729 S) antibodies at R (v. 3.4). After overnight incubation, the immunoprecipitated RNA was washed and purified. cDNA was reverse transcribed from RNA using a High Capacity RNA-to-cDNA Kit (ABI; Cat. # 4387406), and binding targets were quantified by RT-qPCR (Bio-Rad). For the RIP assays with NCI-H660, the cells were harvested and washed with 1X PBS, and the cells were resuspended in 1× PBS and incubated in 1% formaldehyde. Following cross-linking, the cells were pelleted, washed twice with 1× PBS to remove residual formaldehyde, and lysed. The nuclei were pelleted, resuspended, and the chromatin was sheared with sonication. The lysate was added to magnetic beads conjugated to 5 μg of the desired antibody, and the mixture was incubated overnight at 4 °C. The following day the supernatant was removed using a magnetic rack, and the beads were washed five times. The beads were incubated at 70 °C for 1 h to remove crosslinks, followed by adding proteinase K for 30-min incubation at 55 °C to digest proteins. The RNA is removed from the beads and applied to a QIAGEN RNeasy mini kit (Cat No. 74104) for purification. The eluted RNA is converted into cDNA using Invitrogen Superscript IV reverse transcriptase (Cat. No. 18091050). Lastly, qPCR was performed using SYBR green master mix (Roche) for H19 (Forward primer: CAG-GAGTGATGACGGGTGGA, reverse primer: CAGCTGCCACGTCCTGTAA). Successful immunoprecipitation of EZH2 or SUZ12-associated RNA was verified by qRT-PCR using H19 primers with various negative controls including U1snRNA, IGF2/H19 ICR, or Sox2 for validation of on-target and non-target associated RNA.

**Enhanced reduced representation bisulfite (eRRB) sequencing**. The Weill Cornell Medical Center Computational Genomics Core Facility performed enhanced reduced representation bisulfite sequencing. Briefly, bisulfite reads were aligned to the bisulfite-converted hg19 reference genome using Bismark[75]. All samples had bisulfite conversion rates of >99.7%. Percent methylation scores of bisulfite converted cytosines (T's, unmethylated C's) and non-converted Cs (methylated C's) for CpG cytosine methylation were analyzed further using MethylKit (v. 1.11) and R (v. 3.4). Methylation changes for each organoid (OWCM-155-shH19 vs. OWCM-155-Scr) were analyzed separately. Genome-wide differential methylation was calculated between control and shH19 samples. Differentially methylated scores with a cutoff ±25 were chosen as significant. GrCH37/hg19 genome and hg19 CpG sites bed files were downloaded from UCSC and were used as a reference to annotate the differentially methylated regions. ±2-kb upstream and downstream of transcription sites was searched for methylated regions. Custom R scripts were used for file processing and summarizing the output. Bed files were visualized and analyzed using Integrated Genome Viewer (IGV v3.5). Functional association analysis was done utilizing David, and networks were generated using enrich map in Cytoscape.

**Chromatin immunoprecipitation (ChIP) sequencing**. To crosslink proteins to DNA, 540 μl of 37% formaldehyde was added to each 15 cm culture dish containing 20 ml medium for 10 min followed by 2 ml of 10× glycine, swirled briefly to mix, and incubated 5 min at room temperature. Media was removed, and cells were washed two times with 20 ml ice-cold 1× PBS, completely removing wash from culture dish each time. Two millilitre ice-cold PBS (protease inhibitor cocktail) was added to each 15 cm dish. Cells were scraped into a cold buffer. Cells were combined from all culture dishes into one 15 ml conical tube, centrifuged at 2000×g in a benchtop centrifuge for 5 min at 4 °C. The supernatant was removed, and the pellet was stored at −80 °C. ChIP for H3K27me3 and H3K4me3 antibodies was performed by the Chakravati lab at Northwestern University using their established procedures[85], with minor modifications. Cell pellets were resuspended in lysis buffer 1 (50 mM HEPES-KOH pH 7.6,140 mM NaCl, 1 mM EDTA, 10% glycerol, 0.5% IGEPAL-CA630, 0.25%Triton X-100, 1× protease inhibitors) and incubated for 10 min at 4 °C with gentle inversion. Nuclei were recovered by centrifugation (2000×g, 5 min, 4 °C), resuspended in lysis buffer 2 (10 mM Tris-HCl pH 8.0, 200 mM NaCl,1 mM EDTA, 0.5 mM EGTA, 1× protease inhibitors), and extracted for 10 min at 4 °C with gentle inversion. Nuclei were again recovered by centrifugation, resuspended in lysis buffer 3 (10 mM Tris-HCl [pH 8.0], 100 mM NaCl, 1 mM EDTA, 0.5 mM EGTA, 0.1% sodium deoxycholate, 0.5% Sarkosyl, 1× protease inhibitors), and sonicated in an ice-water bath using a Misonix microtip-equipped sonicator at setting 5 (~5 W root mean square output power) for 12 cycles of 15 s on and 45 s off. The sheared chromatin was adjusted to 1% Triton X-100 from a 10% stock solution, and debris was removed by centrifugation at

20,000×$g$ at 4 °C for 20 min. The BCA assay determined the protein concentration of solubilized chromatin. Approximately 700µg of chromatin was immunoprecipitated overnight at 4 °C with antibodies for H3K4me3 (Diagenode, C15410003, 3 µg) and H3K27me3 (Cell Signaling Technologies, 9733, 10 µl). Protein G Dynabeads (30 µl) were added, and immunoprecipitations continued for 3 h. Beads were washed four times with 1 ml of ChIP-RIPA wash buffer (50 mM HEPES-KOH [pH 7.6], 500 mM LiCl, 1 mM EDTA,1.0% IGEPAL-CA630, 0.7% sodium deoxycholate) and once with 10 mM Tris-HCl [pH 8.0], 1 mM EDTA, 50 mM NaCl. The recovered protein–DNA complexes were eluted from the beads twice with 50 µl of 0.1 M NaHCO3, 1% SDS at 65 °C for 15 min with shaking. ChIP and input DNA was adjusted to 0.2 M NaCl, and formaldehyde crosslinks were reversed by heating at 65 °C overnight. DNA was treated sequentially with RNase A and proteinase K and purified using MinElute PCR purification columns (Qiagen). Recovered DNA was quantitated using a Qubit fluorometer (Thermo). ChIP and input DNA libraries were prepared with 5 ng of DNA using KAPA Hyper Prep kits (Kapa Biosystems, KK8502) per manufacturer's instructions and included post-adapter ligation size selection step (0.6×–0.9×) using Ampure XP SPRI magnetic beads (Beckman Coutler, A63881). PCR amplified (11 cycles) libraries were quantitated by Qubit, assessed with a Bioanalyzer (Agilent), and sequenced using 75 bp single-end reads on an Illumina NextSeq 500. Sequencing depths and aligned reads per sample are listed in Supplementary Data 19–20. Data generated were processed by the Epigenomics core at Weill Cornell Medicine. Briefly, Peaks for each replicate ($n = 3$) from each cell line were called from BAM alignment files using MACS2[86] with default parameters. Narrow peaks were called for the H3K4me3 mark, and broad peaks were called for the H3K27me3 mark using the same input for each sample. The peaks were then assessed for coverage and signal distribution using the ChIPQC Bioconductor package and interrogated for peak occupancy and differential binding using the DiffBind R Bioconductor package. For occupancy analysis, Consensus peaks were generated (using the replicates for each mark and cell line) with an overlap rate of 0.66, and bivalent/biphasic regions were defined as regions of overlap between H3K4me3 and H3K27me3 marks for each cell line. Consensus peaks and bivalent/biphasic regions were annotated for proximity to genes using the ChIPseeker R Bioconductor package. Differential binding analysis for V16D/H19 vs. V16D/CTL was performed for the H3K4me3 and H3K27me3 marks. Significantly differentially bound sites for each comparison were then annotated for proximity to genes using the ChIPseeker R Bioconductor package. For GO classification and enrichment, (pAdjustMethod = "BH", pvalueCutoff = 0.05, qvalueCutoff = 0.05) analysis for biological processes, molecular functions, and cellular component was performed using the clusterProfiler Bioconductor R package.

**Sequence and structure conservation**. Multiple sequence alignment (MSA) and phylogenetic analysis were carried out using Ensembls' 'Comparative Genomics' analysis tools and the 'Genomic alignments' analysis feature. *H19* (ENSG00000130600) DNA sequence using human genome HG38 build was used and compared against 69 (70 including human) available whole-genome eutherian ortholog sequences. Species with no alignment in this region were excluded from this analysis ($n = 17$), which resulted in 47 species in total of 70 available within Ensembl v93. In brief and as outlined in their website documentation, the following method was used to produce an MSA and phylogenetic tree for H19. Pairwise whole genome alignments were used to determine conservation, to study the same genomic region in multiple species. LastZ and its predecessor BlastZ are used to align the genome sequences at the DNA level[87,88]. The genomes are compared to one another for comparison between species and to themselves to identify paralogous regions. Whole-genome alignments are the results of post-processing the raw LastZ (or BlastZ) results. Original blocks are chained according to their location in both genomes. The netting process chooses for the reference species (human) the best subchain in each region. These alignments are used to calculate synteny and for scoring orthologue quality. Synteny is defined as the conserved order of aligned genomic blocks between species. It is calculated from the pairwise genome alignments created by Ensembl when both species have a chromosome-level assembly. The search is run in two phases: (1) Search for alignment blocks in the same order in the two genomes. Syntenic alignments that are closer than 200 kb are grouped into a synteny block. (2) Groups in synteny are linked, provided that no more than two non-syntenic groups are found between them, and they are less than 3 Mb apart. A full description of the ortholog genome sequences used, available alignments, phylogenetic synteny calculations, and Ensembl's comparative genomic resources is outlined by Herrero et al.[89]. Other Ensembl resources used in this analysis are described by Zerbino et al.[90], and seen on their website for comparison and analysis of genomes: https://uswest.ensembl.org/info/genome/compara/analyses.html. RNA secondary-structure conservation and arc diagrams for visualization across the 47 of 70 species with sufficient *H19* gene coverage were performed using the R-chie algorithm[91]. The single covariance function was used to estimate covariation in the secondary structure and conservation of 47 species. Coloring of arcs was done based on eight ranges of covariance values (purple to orange), which were calculated based on the base pair covariation range for input MSA[91]. Covariance values ranged from −2.00 (purple: little structure change and high conservation) to 2.00 (orange: high structure change and low conservation). Input parameters included the human H19 secondary structure and the 47-species Ensembl-generated MSA. RNA secondary structure predictions, including MFE calculations, MFE plots, and circle plots, were done using

the mFOLD algorithm[92,93]. Coloring schema for mFOLD's circle plots can be found on their website (http://unafold.rna.albany.edu/www-NAR03/doc/colors.php). The secondary structure used for the conservation analysis with R-chie was the human H19 gap-inserted sequence generated from Ensembl's MSA (described above). All default parameters were selected, as outlined in their paper and webserver (http://unafold.rna.albany.edu/?q=mfold/). Alternative secondary-structure prediction and MFE plots were generated from RNAfold to visualize base pair probabilities integrated within the structure[94]. All default parameters were selected, as outlined in their paper and webserver (http://rna.tbi.univie.ac.at/cgi-bin/RNAWebSuite/RNAfold.cgi).

**RNA Sequence analysis**. We implemented a lncRNA sequence analysis pipeline that includes algorithms catered to detecting known and novel transcripts. Developed in-house, this pipeline is modified and extended from the tuxedo suite of sequence analysis algorithm[28,95]. Once received from the sequencing center in bam format, all sequenced model systems, and patient samples were de-aligned into raw fastq format (including flagged reads) using bam2fastq and put through the pipeline. To ensure high-quality sequence reads, libraries were trimmed using a windowed-adaptive approach (Sickle – https://github.com/ucdavis-bioinformatics/sickle). The algorithm determines the most optimal inner read sequence for each read pair processed together by trimming both 3′ and 5′ prime ends based on quality and length thresholds (for full description, see—http://bioinformatics.ucdavis.edu/software/). Bases with a quality score of less than 99.0% base call accuracy (corresponding to a Phred quality score of 20) were removed. Reads less than ~2/3 read length (30nt in WCM and 60nt in VPC) post-trimming were discarded. Highly repetitive sequences (>2% of library) were also discarded post-trimming using the cutadapt tool. All quality control metrics were generated and quantified (pre- and post-trimming) using FASTX-Toolkit and FastQC software. Reads were aligned to the Hg38 human genome build using an unspliced aligner for handling exonic reads (Bowtie - v2.2.3), in conjunction with a spliced aligner to handle reads spanning exon-exon junctions (Tophat—2.0.12). Transcriptome reconstruction using Ensembl v86 gene tracks and Human genome build GRCh38 for each library was performed using a quasi de novo (genome-guided) approach (Cufflinks—v2.2.1), where reads were assembled, and abundances estimated using an overlap graph producing a minimal spanning network of transcripts. With this isoform-aware approach (Cufflinks), alternative isoforms or transcript variants can be identified and quantified. This version of Ensembl contained 38 transcript classes grouped by four core biotypes. At this stage, transcripts were also multi-read and fragment bias-corrected. Transcripts with highly abundant expression were masked (e.g., rRNAs) from downstream steps to increase transcript quantification accuracy. Sample transcriptomes, the reference genome, and the transcript annotation were then meta-assembled (Cuffmerge) to produce a single annotation transcriptome model. Based on this model, transcript quantification (Cuffquant) and normalization (Cuffnorm), considering varying library depths and transcript lengths, were performed. Transcript expression displaying computational artifacts (expression values < 0.1 known to occur with Cufflinks) were converted to zero values. All algorithms denoted in brackets are referenced and previously described[95].

**Statistical analysis, reproducibility, and data representation**. Data represented were performed in ≥3 independent experiments or biological replicates. Statistical analysis of changes was performed by unpaired Student's t-tests or two-way ANOVA (Tukey's or Sidak's multiple comparison tests) as noted. Significance was represented by *$p < 0.05$; **$p < 0.01$; ***$p < 0.001$, and ****$p < 0.0001$ unless specifically noted. Reproducibility was ensured for all the representative WBs and micrograph images by repeating the experiment in 3 different biologically independent conditions. TF binding sites for H19 were identified through TomTom and Jasper algorithms (Supplementary Data 4–6). The programming language R v3.0 was used for statistical analysis. Unsupervised hierarchical clustering was performed with the h.clust package with Pearson correlation for distance and average linkage used. The clustering and heatmaps generated were built using the heatmap.2 function. Similar clustering analysis was performed for GRID cohorts except with Euclidian distance, the ward method for linkage, and the use of the heatmap.3 function due to its advanced row/column labeling features. Normalized log2 expression values were standardized/scaled using a $Z$-score that ranged from −2 to 2. For principal component analysis, the R package prcomp was used to calculate variance among transcript and sample subsets for the calculation of transcript weights and principal components. The top three components were used for visual inspection. Receiver–operating characteristic (ROC) curves and area under the curve[96] calculations, the R package "pROC" was used. Kaplan–Meier analysis was performed for determining survival outcome using the R package "survfit" with transcripts displaying below background (<0.1) expression being removed from this analysis for microarray profiled cohorts MCI and MCII.

**Reporting summary**. Further information on research design is available in the Nature Research Reporting Summary linked to this article.

# Data availability
All clinical patient sequencing and microarray data (Table 1) were available in-house through previous publication submissions. Initial description, interrogation, and results

for these datasets are available in referenced publications. Please see methods "Clinical patient samples" for publication references. For access to these datasets, please use the following accession codes in reference to cohort labels from Table 1: VPC (PRJEB19256, PRJEB21092, PRJEB6530, PRJEB9660), BCCA (EGAD00001004139/EGAC00001000914), MCI (GSE46691), MCII (GSE62116), and JHMI (GSE104786). WCM1, WCM2, WCDT, and GRID cohorts require original study author permission and are restricted access patient cohorts. Sequencing performed in this study was deposed under accession codes GSE182914 (eRRB—Methylation Sequencing—https://www.ncbi.nlm.nih.gov/geo/query/acc.cgi?acc=GSE182914) and GSE183983 (ChIP Sequencing—https://www.ncbi.nlm.nih.gov/geo/query/acc.cgi?acc=GSE183983). The source data for Figs. 3I, K, 4H, I, K, 5D, 6B, 6C, D, F are provided as a Source Data file. All the other data are available within the article and its Supplementary Information. Source data are provided with this paper.

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

## Acknowledgements

This study was supported by the National Cancer Institute (P30 CA023074, SPORE P50-CA211024 to H.B.), Department of Defense (PCRP W81XWH-18-1-0533 to N.S., PCRP W81XWH-13-1 to H.B., W81XWH-12-1-0560 A.S.K.), Mitacs Accelerate Ph.D. Fellowship Program (IT04310 to V.R.R.) in collaboration with Decipher Biosciences, National Institute of Health (R35CA232105 to F.J.S., R37CA241486 to H.B., R01CA173200 to A.S.K.), Prostate Cancer Foundation (H.B.), Terry Fox Foundation (TFRI NF PPG Project #1062 to C.C. and M.G.), Prostate Cancer Canada Team Grant (T2013-01 to C.C.), Canadian Foundation of Innovation—Innovation Fund (33440 to C.C.), Canadian Institutes of Health Research (PJT-153073 and PJT-175238 to C.C.), and Prostate Cancer Foundation British Columbia Grant-in-Aid (V.R.R.). A.S.K. was supported by the Lauder Foundation through Dr. David Alberts. The authors acknowledge the Experimental Mouse Shared Resource for helping with in vivo experiments. We are very grateful to Dr. Carolyn J. Brown for her advice surrounding theories of H19 mechanism, function, and sequence polymorphisms. We are also extremely grateful to Stephanie Giles Ramnarine for her manuscript comments, advice, and support.

## Author contributions

Conception and design: N.S., V.R.R., V.O., C.C., and A.S.K; Development of methodology: N.S., V.R.R., J.H.S., D.M., M.M.H., V.O., C.C., and A.S.K; Acquisition of data: N.S., V.R.R., J.H.S., M.N., M.K., V.O., S.K.R.P., K.O., D.M., M.M.H., B.S., and J.B.P.; Analysis and interpretation of data: N.S., V.R.R., J.H.S., R.P., M.N., M.K., P.M., D.M., M.M.H., J.B.P., B.S., M.Z., J.J.B., K.O., S.K.R.P., V.O., C.C., and A.S.K.; Writing/review of the paper: N.S., V.R.R., J.H.S., R.P., M.N., M.K., B.R.L., M.M.H., D.M., P.M., J.J.B., K.O., S.K.R.P., E.A.G., M.A., R.J.K., R.A., R.B., A.Z., M.G., E.D., H.B., V.O., F.J.S., C.C., and A.S.K., Administrative, technical, or material support: N.A.W., B.R.L., J.J.B., K.O., S.K.R.P., R.J.K., D.C., E.J.S., R.A., F.F., Y.W., R.B., A.Z., M.G., E.D., M.R., H.B., C.C., and A.S.K.; Study supervision: F.J.S., C.C., and A.S.K.

## Competing interests

E.A. Gibb and E. Davicioni are employees of Decipher Biosciences, Inc. Michelle M. Hanna and David McCarthy are employees of Ribomed Biotechnologies, Inc. The remaining authors declare no competing interests.

## Additional information

[1]University of Arizona Cancer Center, University of Arizona, Tucson, AZ 85724, USA. [2]Vancouver Prostate Centre & Department of Urologic Sciences, University of British Columbia, Vancouver, BC, Canada. [3]Harvard Medical School Initiative for RNA Medicine, Harvard Medical School, Boston, MA 02215, USA. [4]Department of Cellular and Molecular Medicine, University of Arizona, Tucson, AZ 85724, USA. [5]Department of Molecular Biology and Biophysics, UConn Health Center, Farmington, CT 06030, USA. [6]Department of Medicine, University of Arizona, Tucson, AZ 85724, USA. [7]Department of Physiology, University of Arizona, Tucson, AZ 85724, USA. [8]Ribomed Biotechnologies, Inc., 8821N. 7th St. STE 300, Phoenix, AZ 85020, USA. [9]Epigenomics Core Facility, Weill Cornell Medicine, New York, NY 10065, USA. [10]Department of Obstetrics and Gynecology, Division of Reproductive Science in Medicine, Feinberg School of Medicine, Northwestern University, Chicago, IL 60611, USA. [11]Decipher Biosciences, Inc, Vancouver, BC, Canada. [12]Department of Radiation Oncology, University of California San Francisco, San Francisco, CA 94158, USA. [13]Department of Urology, Mayo Clinic College of Medicine, Rochester, MN 55905, USA. [14]Department for BioMedical Research, University of Bern, 3008 Bern, Switzerland. [15]Department of Medical Oncology, Dana-Farber Cancer Institute, 44 Binney St, Boston, MA 02115, USA. [16]These authors contributed equally: Neha Singh, Varune R. Ramnarine. ✉email: colin.collins@ubc.ca; akraft@uacc.arizona.edu

