## [Peer Review File · Nature Communications]

The long noncoding RNA H19 regulates tumor plasticity in neuroendocrine prostate cancerReviewers' comments:

Reviewer #1 (Remarks to the Author):

In this manuscript, Singh and colleagues provide evidence that the lncRNA H19 regulates tumor plasticity in NEPC. Authors first showed increased H19 expression in multiple NEPC datasets and NEPC models. Functional knockdown and overexpression experiments implicate this lncRNA in NEPC. They further provided some evidence that H19 interacts with EZH2/PRC2 complex and regulates global methylation. Finally, the authors suggest H19 expression levels as a potential predictive and prognostic biomarker for NEPC. Overall, the bioinformatics-based exploration and data associating increased H19 with NEPC are solid. However, there lack in vivo functional tumor studies and many mechanistic aspects are descriptive and weak.

MAJOR POINTS:

1. Regarding a causal role of H19 in NEPC: ADT or other therapy-induced 'trans-differentiation' of adenocarcinomas to CRPC-NE resembles iPSC reprogramming induced by chemicals and involves stepwise changes in the epigenome leading to alterations of key gene expression profiles and phenotypic changes in the cells. In this process, the AR+ adenocarcinomas will undergo temporal transition to AR-/lo NE-like tumors. The role of H19 in this process remains undefined. In other words, is H19 involved in the initiation of NEPC transition or more involved in the maturation and maintenance of NEPC phenotypes? It's encouraging that H19 expression levels positively correlate with MYCN, SOX2, and EZH2 (key regulators of neurogenesis and neural development) and with terminal NEPC differentiation markers CHGA/CHGB/SYP. Experiments need to be designed (e.g., inducible system) to address whether H19 induction is both sufficient and required for NEPC initiation, establishment and/or maintenance. Importantly, there lack in vivo tumor studies to show the importance of H19 in mediating and supporting NEPC growth.
2. Regarding H19 and AR/AR signaling: Along with the above point, presumably H19 will have to shut down AR expression/signaling prior to the emergence of NE phenotype? This should be directly tested and underlying molecular mechanisms provided. Then the questions become: How does H19 induction repress AR expression (e.g., Figure 3E)? Does H19 directly interact with AR? These and other related questions are important because the answers will help determine whether the H19-involved NEPC reprogramming represent a direct effect of H19 or via a 'default' pathway (i.e., due to repression of AR/AR signaling)? Related to point, author should show the changes of AR and PSA levels in most H19-manipulated cells/models (e.g., Fig. 4E). Vice versa, how exactly AR signaling regulate H19 expression (Supplementary figure S8E-F)?
3. Related to above and regarding H19 and sensitivity of cells to enzalutamide (Enza):
 - a. What are the relative expression levels of H19 in LNCaP-WT, LNCaP-shP53/RB1, LNCaP-SL cells.
 - b. Fig. 4F suggests that Enza does not work in AR+H19+ cells. As results showed that H19 drives NE feature (Fig. 3) and the negative correlation between AR and H19 (Fig. S8E), what is the minimum level of H19 expression to make LNCaP cells indifferent to Enza treatment? Are there any changes in AR mRNA or protein level along with addition of Enza in LNCaP-H19 cells (Fig. 4F)?
4. Regarding H19 upregulation in NEPC: Along with the above point in #3, most imprinted genes are regulated via methylation/demethylation in the DMR (Differentially Methylated Region). Although authors here implied the upregulation of H19 during NEPC transition by stemness factors (e.g., SOX2, OCT4, NANOG), no direct evidence is provided. And there is no information on whether H19 upregulation involves methylation/demethylation in the H19-DMR during NEPC reprogramming.
5. Regarding H19/IGF2 imprinted locus in NEPC reprogramming: Like several other developmentally regulated imprinting locus, the maternal imprinting at H19/IGF2 locus has been shown to help maintain the adult HSC quiescence (Nature, 500:345-9, 2013). What's the imprinting status of this locus and what's the IGF2 expression status in de novo NEPC and during ADT/therapy-induced NEPC transition?
6. Regarding H19-derived miR-675: Much of the H19 functions in long-term quiescent HSCs is mediated via the H19-derived miR-675, which targets the IGF-1R (Nature, 500:345-9, 2013). It's unclear whether the reported H19 effects here in NEPC involve miR-675.
7. Regarding 'Stem cell' and NE genes: Authors have frequently shown how manipulations of H19 lead to changes in the stem cell vs. NE genes but the relationship between these two classes of genes remains unclear. For example, does ADT/Enza treatments first upregulate stemness genes, which then turn on NE genes to promote NE phenotype? More importantly, there lacks a clear

understanding exactly how H19 'control' the expression of these genes. Authors might pick one gene from each class to conduct a more in-depth investigation. Vice versa, it's completely unclear how any of the stemness factors actually control H19 expression (Supplementary figure S7G).

8. Regarding H19 and EZH2/PRC2: It seems that MANY lncRNAs including H19 would bind EZH2/PRC2 complex but the binding specificity remains obscure in most cases. In the current context, although authors provided nice evidence for the high preservation of the H19 primary sequence as well as secondary structures across the species (Figure 2 and Supplementary figure S3-S5), there is no information on which part of H19 actually binds to which domain(s) of EZH2. Do both sense and anti-sense strand of H19 bind EZH2 and come in contact with the PRC2 complex? Along this line, it's unclear exactly how H19-EZH2 interactions lead to a transition to the NE phenotype?

9. About expression levels of H19 from RNA-seq and RT-PCR: To compare the overall relative expression levels of H19 in different cohorts and samples/models/contexts, authors should consider presenting the original Ct values of H19 and the normalized read counts or TPM (if possible) in different models (including clinical samples, cell lines, mouse models and organoids) in a Supplementary table.

10. About box plots: In all box plots throughout the paper, authors should present the n (patient/case numbers) and other parameters such as centerlines and whiskers.

11. About H19 and alteration in DNA methylation:

a. As shown in Fig. 3, knocking down of H19 turns off NE genes and turns on luminal genes. What are the changes of methylation status on luminal markers such as CK8, CK18 and NKX3.1 and NE markers.

b. The genes listed in Fig. 6C are not consistent with those in Fig. 6D and Fig. 6E. For example, NKX2.1 in Fig. 6D is not shown in Fig. 6C, while BCL11B and CCND2 in Fig. 6E were not presented in Fig. 6D and 6C.

12. Fig. S1 (top): Is it VPC-P or VPC-M? In this table, p-values should be denoted with * $p < 0.05$, ** $p < 0.01$, and *** $p < 0.001$. As presented, the data can be confusing and misleading. For instance, SOX11 does not seem to differ in the WCDT, SOX9 and many other genes exhibit divergent and opposite changes but authors only vaguely stated ".....(Supplementary figure S1, p-value < 0.05 unless denoted by *)." (page 7, end of the top paragraph). Importantly, INCONSISTENT patterns (ups vs. downs) of most genes analyzed across the 5 datasets should be discussed.

MINOR POINTS:

1. Ref. 28 and 40 are the same.
2. Supplementary figure S2, as presented, is not very informative, as there are multiple 'neuron', 'neural', 'neurogenesis' gene expression terms that authors should interrogate. Also, the NES, FDR, and p-value should be presented in the GSEA plot.
3. Fig. 1D has 38 genes instead of 37 genes mentioned in the Text (page 7; second to last line).
4. In Supplementary figure S8E, what's the unit of H19 expression? The Y-axis presentation should start from "0". In figure S8F, the "+" should be "-" in the absence/presence of DHT.
5. Supplementary figure S3: n and p-values should be indicated directly in the box plots. The Y-axis unit should be indicated.
6. How are the sensitivity score and specificity score calculated?
7. Page 9, third line from the bottom of the top paragraph: 'Figure 4D-H10i....' should be 'Figure 2D....'.
8. Figure 6B and Figure 6C were mis-cited in the text (page 13) which did not match the actual data.
9. There are also many other typographical and grammatical errors.

Reviewer #2 (Remarks to the Author):

This manuscript concerns the functional role of the lncRNA H19 in lineage plasticity and NEPC differentiation. H19 expression increases in treatment induced NEPC relative to CRPC (which is AR dependent) and also may be higher in primary NEPC relative to adenocarcinoma. Using various model systems, the authors also suggest that H19 regulates NE and stem cell genes as well as

proliferation and invasion. H19 has previously been found to bind EZH2, which the authors demonstrate occurs in prostate cancer cells and also show here that exogenously modulated H19 is correlated with inverse changes in total H3K27me3 levels. In addition, depletion of H19 in a NEPC cell line resulted in a changed DNA methylation pattern that partially overlapped (~10%) with NEPC/CRPC dichotomy found in clinical samples.

The concept that H19 is a functional component of epigenetic programming associated with adenocarcinoma to neuroendocrine commitment is of interest. At this time, I would consider the data quite preliminary in validating the hypothesis. The analysis of the model systems is incomplete and misinformed in a few instances. Although the mechanistic analyses are a good start, they lack the depth and completeness necessary to draw conclusions concerning mechanisms of action. Finally, the suggestion that H19 may be diagnostically useful relies on a limited number of descriptive cases. A more complete discussion follows.

Figure 1 and bioinformatics analyses: H19 expression appears clearly higher in NEPC than adenocarcinoma (AD). Of particular interest would be how H19 expression varies across CRPC phenotypes. In the WCM-2 cohort, for example, it would be of interest to analyze the available AR signature scores and NE signature scores vs. H19 expression across the CRPC cohort. This type of analysis would be more informative for deciphering a general role in dedifferentiation/lineage plasticity/NE commitment than comparing primary adenocarcinoma to treatment induced NEPC (e.g. Figure 1C). The procedure used for the selection of samples in Figure 1D is not clear as AD samples from some cohorts (e.g. WCM-1 and WCDT) appear to have been left out of the analysis.

Figures 3: The data in these figures reflect either ectopic expression or knockdown of H19 in a variety of NEPC cell lines derived from patients, experimental human cell lines, or primary human prostate or mouse model organoids. Organoids can be quite sensitive to off-target effects of shRNAs, and therefore additional shH19 constructs should be compared or preferably, rescue experiments should be performed. Most often, relative RNA expression for a few specific genes was compared within the total population. Such data is not clearly interpretable because the actual level of RNA expression is not indicated and could represent a very modest fraction of cells or a physiologically insignificant level of expression. Much more informative and convincing would be histological staining of relevant markers in fixed cell populations or molecular single cell analyses. Organoids from primary tissues are highly heterogeneous, and the methodology of deletion used here did not distinguish the various cell types. For example, other investigators do not find significant histological/molecular NE differentiation for in vitro deleted Tp53/Rb1 mouse prostate organoids (Fig. 1J) or in Tp53/Pten null mouse models (Fig S6F). Also, the predominant growth of normal basal organoids from primary human prostate cancer prostatectomy specimens has been observed by many researchers. Therefore, it is incumbent upon the investigators here to demonstrate that Hu-AdPC organoids (Figs. 3E and 4G) are in fact luminal prostate cancer.

Figure 4: This figure addresses growth regulation by H19 depletion in NEPC models as well as the suggestion that H19 "re-sensitizes" NE models to androgen antagonism. The comments above about the appropriate characterization of the Tp53/Rb1 null mouse model system and the specificity of shH19 methods apply here. Organoids are quite sensitive to non-specific toxic effects of lentivirus infection. Titrating lentivirus relative to growth inhibition and protein expression is necessary for conclusive organoid growth assays (see Fig 4A). Relevant to growth inhibition, in Fig. 4E, it appears that ENZA treatment and shH19 may be simply additive and not suggestive of a gain in ENZA sensitivity. The concept that H19 loss leads to AR dependence is an interesting and important one that requires determining AR protein expression levels as well as analyzing multiple AR dependent genes.

Figures 5 and 6: The modulation of total H3K27Me3 relative to H19 levels and the association of H19 with EZH2 and SUZ12 does not show H19-mediated regulation of the PRC2 complex and epigenetic regulation of NEPC genes. This requires specific analyses of H19/PRC2 occupancy and function at selected gene regulatory regions. In addition, DNA methylation studies are at best a highly indirect measure of PRC2 activity.

Reviewer #3 (Remarks to the Author):

In their manuscript titled, "The long noncoding RNA H19 regulates tumor plasticity in neuroendocrine prostate cancer," Sigh N, Ramnarine VR and colleagues suggest a previously unrecognized role for H19 in regulation of the prostate cancer neuroendocrine phenotype. The work is interesting and contains many novel elements, and the manuscript is well-written. I have the following comments:

- 1) The well-known function of H19 is in the regulation of the growth factor IGF2. H19 is often silenced through loss of imprinting in human cancers, thereby relieving the suppressive effect on IGF2 expression. Demethylating agents are known to induce re-expression of H19, suppress IGF2 function, and thereby inhibit cancer growth. This well known function is not explored in the manuscript, and it is difficult to contextualize the findings without an exploration of this well-established H19 signaling axis.
- 2) Minor comment: It is unclear what the authors mean by the term "clinically recurrent" (Page 8) in the context of dysregulated gene expression.
- 3) The conservation analyses in Figure 2 are interesting, but not entirely novel, and unclear how they support the specific message in the paper. The fact that H19 is a highly conserved noncoding RNA has been known in the field for a long time (see e.g. Smits G et al, Nat Genet, 2008). Conservation of the secondary structure of H19 has also been well-established.
- 4) It is unclear why the authors chose not to consider H19ii as a relevant isoform in NEPC. They only state that it is not reported within GENCODE or RefSeq – but this does not mean it is not relevant.
- 5) Minor comment: I may have missed it, but what controls were used to assess whether the RNA immunoprecipitation experiments did not simply detect binding of polycomb complex members to the genomic locus encoding H19?

Response to reviewers' comments:

We would like to express our gratitude for the time and effort of each reviewer and their constructive criticism and feedback. Given recent events, we have addressed these comments providing additional mechanistic insights and further evidence supporting our results. We have extensively revised the initial submission including new results and observations with the addition of new ChIP-sequencing, ChIP-qPCR, promoter luciferase assays, immunohistochemistry, *in vivo* experiments, highly sensitive methylation detection on *IGF2/H19* ICR in AdPC and NEPC, and computational interrogation of deeper isoform identification and quantification, correlation of expression to known public NEPC/AR signature scores in clinical specimens, and transcription factor binding site identification. Collectively, this represents a significant amount of new data, which responds to reviewer comments and strengthens our manuscript. In addition to this new data, we have also improved the quality of our original data increasing readability and interpretation. Taken together, these data support the critical role that H19 plays in NEPC transdifferentiation (NEtD) and NEPC. We hope that the revised manuscript will be considered for publication.

Below each reviewer comment is our response written **in red**. New additions in the main manuscript text have been written in **Blue**.

Reviewer #1 (Remarks to the Author):

In this manuscript, Singh and colleagues provide evidence that the lncRNA H19 regulates tumor plasticity in NEPC. Authors first showed increased H19 expression in multiple NEPC datasets and NEPC models. Functional knockdown and overexpression experiments implicate this lncRNA in NEPC. They further provided some evidence that H19 interacts with EZH2/PRC2 complex and regulates global methylation. Finally, the authors suggest H19 expression levels as a potential predictive and prognostic biomarker for NEPC. Overall, the bioinformatics-based exploration and data associating increased H19 with NEPC are solid. However, there lack *in vivo* functional tumor studies and many mechanistic aspects are descriptive and weak.

MAJOR POINTS:

1. Regarding a causal role of H19 in NEPC: ADT or other therapy-induced 'trans-differentiation' of adenocarcinomas to CRPC-NE resembles iPSC reprogramming induced by chemicals and involves stepwise changes in the epigenome leading to alterations of key gene expression profiles and phenotypic changes in the cells. In this process, the AR+ adenocarcinomas will

undergo temporal transition to AR-/lo NE-like tumors. The role of H19 in this process remains undefined. In other words, is H19 involved in the initiation of NEPC transition or more involved in the maturation and maintenance of NEPC phenotypes? It's encouraging that H19 expression levels positively correlate with MYCN, SOX2, and EZH2 (key regulators of neurogenesis and neural development) and with terminal NEPC differentiation markers CHGA/CHGB/SYP. Experiments need to be designed (e.g., inducible system) to address whether H19 induction is both sufficient and required for NEPC initiation, establishment and/or maintenance. Importantly, there lack *in vivo* tumor studies to show the importance of H19 in mediating and supporting NEPC growth.

We have further documented the role of H19 in transdifferentiation process by utilizing multiple biological systems. First, to approach this question in an unbiased manner, the RNA levels of H19 were measured in LTL 331 PDX, a model for neuroendocrine (NE) transdifferentiation (NEtD). In this model (PMID 26235627), tumors are injected subcutaneously in mice and as they grow under androgen deprivation conditions they transdifferentiate to NEPC. We removed these tumors at various time points during differentiation and isolated and sequenced the RNA. Examining the RNA expression, we observed an increase in H19 transcripts during both early (post-castration) and late (NE development) stages of transdifferentiation (NEtD), suggesting a role for H19 in both the initiation and maintenance of the NEPC phenotype (new Fig. 5A). These findings have been added on page 13 of the revised manuscript. Second, we have evaluated LNCaP cells cultured under prolonged androgen independent conditions (40 days) in which these cells transdifferentiate to NEPC, demonstrating elevated H19 levels (new Fig. 5D). These findings have been added on page 13 of the revised manuscript. Third, as suggested, we designed a doxycycline (DOX) - inducible vector for H19 overexpression. We have stably expressed this vector in 2 cell lines- LNCaP and C4-2B and found that the induction of H19 expression with DOX increased the level of SOX2, EZH2 and terminal NE markers at both the RNA and the protein level (new Fig. 3F, 6F). These findings have been added on page 11 and 15 of the revised manuscript. Finally, we performed *in vivo* experiments using human NEPC organoids transduced with shH19 or control vector. Analysis of this experiment demonstrated that knockdown of H19 inhibited NEPC growth and maintenance of the NEPC phenotype (new Fig. 4D-F). These findings have been added on page 12 of the revised manuscript. Together these experiments suggest an important role for H19 in NEPC. Additional mechanistic details addressing these concerns are also provided in the responses to the critique found below.

- Regarding H19 and AR/AR signaling: Along with the above point, presumably H19 will have to shut down AR expression/signaling prior to the emergence of NE phenotype? This should be directly tested and underlying molecular mechanisms provided. Then the questions become: How does H19 induction repress AR expression (e.g., Figure 3E)? Does H19 directly interact with AR? These and other related questions are important because the answers will help determine whether the H19-involved NEPC reprogramming represent a direct effect of H19 or via a 'default' pathway (i.e., due to repression of AR/AR signaling)? Related to point, author should show the changes of AR and PSA levels in most H19-manipulated cells/models (e.g., Fig. 4E). Vice versa, how exactly AR signaling regulate H19 expression (Supplementary figure S8E-F)?

We have consistently observed an inverse correlation between H19 and the AR signature genes (including AR downstream signaling genes PSA, KLK2, KLK4, NKX3.1) (new Fig. 1D). However, we did not find that changes in H19 expression caused any major changes in AR protein expression. RNA:protein cross linking experiments done in V16D/H19 cells did not show direct binding of AR to H19 (Fig. 1 included in this response below). As a control for this experiment, lncRNA HOTAIR did bind AR as previously reported (PMID 26411689).

Figure1: RNA Immunoprecipitation experiments indication that H19 did not bind to AR protein. HOTAIR was used as positive control to demonstrate the AR binding.

Also, as demonstrated using cycloheximide chase experiments (Fig. 2 included in this response below), there was no change in the protein half-life of AR with H19 overexpression.

Figure 2: Cycloheximide chase experiment demonstrating that H19 overexpression did not alter the stability of AR protein. Orange curve: V16D-Control, Blue curve: V16D-H19.

Although there were no major changes in AR protein, ChIP-seq done in V16D cells overexpressing H19 showed that H3K4me3 transcriptionally active histone marks were reduced in the promoter of the AR and genes that are regulated by the AR. In this experiment, there were distinct changes in the promoter of androgen responsive genes, e.g. KLK2, KLK4, NKX3.1 (new Fig. 7D-E), in H3K4me3 levels. These results (see page 15 of the revised manuscript) suggest that H19 epigenetically regulated the AR signaling.

We have further investigated how androgen regulates H19 transcription. The addition of dihydrotestosterone (DHT) to LNCaP and C4-2B cells suppressed H19 expression (new Fig. 5B). While the addition of enzalutamide to these cells stimulated increases in H19 expression (new Fig. 5C). Consistent with these results, the addition of DHT reduced the read out of a luciferase reporter when driven by the H19 promoter. Enzalutamide (ENZA) addition did the opposite (new Fig. 5G). These results (page 13-14 of the revised manuscript) suggested that androgen is transcriptionally regulating the H19 gene.

To examine the mechanism of H19 promoter regulation by the AR, we performed chromatin immunoprecipitation (ChIP) with AR antibody using C4-2B cells treated with DHT or enzalutamide. The potential ARE binding regions in the H19 upstream region were identified by using the JASPER database and modelling (PMID 14681366, PMID 31701148) (new Supplementary table 6, Supplementary figure S6E). ChIP qPCR confirmed 4 ARE binding motifs

with enhanced AR enrichment upon DHT treatment (new Supplementary Fig. S12D). ENZA treatment of the C4-2B cells reduced the binding of AR to these 4 sites (new Fig. 5F). We postulate that the binding of AR to these sites suppressed H19 transcription which is reversed by enzalutamide treatment. These findings (page 14 of the revised manuscript) demonstrate that androgen deprivation during NEtD played a role in driving H19 transcription.

3. Related to above and regarding H19 and sensitivity of cells to enzalutamide (Enza):
 - a. What are the relative expression levels of H19 in LNCaP-WT, LNCaP-shP53/RB1, LNCaP-SL cells.

The relative expression levels of H19 in LNCaP shTP53/Rb1 as compared to LNCaP shSCR (control) is 2-fold and in LNCaP-SL (stem-like) when compared to parental LNCaP cells (control) is 3.5 fold. (new Supplementary figure S8F, S8I respectively).

- b. Fig. 4F suggests that Enza does not work in AR+H19+ cells. As results showed that H19 drives NE feature (Fig. 3) and the negative correlation between AR and H19 (Fig. S8E), what is the minimum level of H19 expression to make LNCaP cells indifferent to Enza treatment? Are there any changes in AR mRNA or protein level along with addition of Enza in LNCaP-H19 cells (Fig. 4F)?

To get a sense of the level of H19 needed to regulate NEPC differentiation, we have compared the level of this lncRNA in V16D cells (androgen resistant cells derived from LNCaP cells) and 42D which are NEPC-like and derived from V16D. 42D had a 12-fold higher H19 transcript level and were resistant to enzalutamide (new Supplementary Fig. S6C). Comparing LNCaP (adenocarcinoma) cells to NEPC cells the H19 level is increased approximately 30-40 fold (Fig. 3B). We suspect that the increase in H19 needed to induce NEPC is highly dependent on cellular context and the level of AR signaling including exposure to enzalutamide or other inhibitors.

Although we did not see changes in AR levels with H19 overexpression or knockdown alone, we did find that when H19 levels changed after ENZA addition for 72h. H19 had an impact on AR protein levels (see quantification in new Fig. 4K) but only when ENZA was present.

4. Regarding H19 upregulation in NEPC: Along with the above point in #3, most imprinted genes are regulated via methylation/demethylation in the DMR (Differentially Methylated Region). Although authors here implied the upregulation of H19 during NEPC transition by stemness factors (e.g., SOX2, OCT4, NANOG), no direct evidence is provided. And there is no

information on whether H19 upregulation involves methylation/demethylation in the H19-DMR during NEPC reprogramming.

We have collaborated with Ribomed Biotechnologies, Inc., a company focused on bisulfite free sensitive DNA methylation profiling analysis, to experimentally investigate the methylation/demethylation in the ICR region of H19. A sensitive bisulfite free assay (Methylmeter assay; PMID 27337298), of the methylation level of the CpG island in the imprinting center just upstream of the H19 promoter was performed (Supplementary Methods). We probed the 1624 bp MseI fragment containing the CTCF binding regions 1-3 which is normally methylated on the paternal copy with imprinting. DNA samples from cell lines and controls were cleaved with MseI and the numbers of methylated versus unmethylated DNA copies in the 2 fractions (Supplementary figure S8D) are quantified by Coupled Absorption PCR signaling (CAPS) to express the percent methylation (Supplementary Methods, Supplementary figure S6D). Results indicated that prostate cancer (PCa) cell lines (LNCaP, C4-2B and V16D) had a higher percent of ICR methylation as compared to NEPC cell line (NCI-H660) and organoid (OWCM-1262) which are less methylated and are within the normal imprinting range (40-60% methylation) (new Figure 3C). It appears that there is an increase in H19 methylation early in cancer development (biallelic silencing). Transition to NEPC is associated with a decrease in H19 methylation and a return to “normal” imprinting levels (new Fig. 3C). These findings have been added to page 10-11 of the revised manuscript.

New data in C4-2B cells showed that androgen depletion elevated SOX2 levels, and this transcription factor drove increases in H19 levels (new Supplementary fig. S12E). Analysis of SOX2 binding regions in the proximal H19 promoter demonstrated several potential binding sites (new Fig. 5I, top). Examining C4-2B cells treated with Enzalutamide (10 μ M, 6days), ChIP with anti-SOX2 antibody showed enrichment of SOX2 binding on 2 specific sites as compared to (new Fig. 5I, bottom) control. Mutating one of the SOX2 sites on the H19 promoter (new Fig. 5J, top) caused a significant reduction in ENZA induced H19 promoter luciferase activity (new Fig. 5J, bottom). Together these results (page 13-14 of the revised manuscript) indicate the transcription of H19 appears to be regulated in part by the binding of SOX2 whose level is markedly increased during androgen deprivation and consequent NE transdifferentiation.

5. Regarding H19/IGF2 imprinted locus in NEPC reprogramming: Like several other developmentally regulated imprinting locus, the maternal imprinting at H19/IGF2 locus has been shown to help maintain the adult HSC quiescence (Nature, 500:345-9, 2013). What's the imprinting status of this locus and what's the IGF2 expression status in de novo NEPC and during ADT/therapy-induced NEPC transition?

IGF2 expression across cell lines and organoid systems ranging from Normal→AdPC→NEPC (including de novo (NCI-H660) and t-NEPC) (new Supplementary Fig. 14A) indicate that expression is slightly elevated in the NEPC. This increase is significantly less than that observed in H19 expression (new Fig. 3A-B). The methylation of *IGF2/H19* ICR in AdPC and CRPC is elevated while NEPC cells were found to be hypomethylated in comparison demonstrating normal imprinting levels (new Fig. 3C). These results are briefly discussed on page 19 of the revised manuscript.

6. Regarding H19-derived miR-675: Much of the H19 functions in long-term quiescent HSCs is mediated via the H19-derived miR-675, which targets the IGF-1R (Nature, 500:345-9, 2013). It's unclear whether the reported H19 effects here in NEPC involve miR-675.

To explore the role of miR-675 in NEPC and NEtD, transduction of a construct containing a portion of the H19 sequence expressing mir675 in a murine model failed to induce NEPC markers (new Supplementary Fig. 14C-D). These results (briefly discussed on page 18 of the revised manuscript) suggested that miR-675 is not involved in H19 mediated NEPC induction.

7. Regarding 'Stem cell' and NE genes: Authors have frequently shown how manipulations of H19 lead to changes in the stem cell vs. NE genes but the relationship between these two classes of genes remains unclear. For example, does ADT/Enza treatments first upregulate stemness genes, which then turn on NE genes to promote NE phenotype? More importantly, there lacks a clear understanding exactly how H19 'control' the expression of these genes. Authors might pick one gene from each class to conduct a more in-depth investigation. Vice versa, it's completely unclear how any of the stemness factors actually control H19 expression (Supplementary figure S7G).

The temporal status of NEPC and stem cell gene expression has been studied previously by our team (PMID 26235627). In this article, the authors showed an increase in stem cell gene expression prior to the elevation in NEPC gene expression during the transdifferentiation process. Bioinformatic analysis of this data for H19 expression showed increased H19 levels occurring prior to elevation of SOX2 during the transdifferentiation (NEtD) in the LTL331R model system (new Fig. 5A, new Supplementary fig. S11), suggesting that androgen deprivation could play a role in controlling H19 levels. These findings are added to page 13 of the revised manuscript.

To further investigate the regulation of NEPC genes by H19 we performed chromatin immunoprecipitation sequencing (ChIP-seq) in V16D/CTL (control vector transduced) and V16D/H19 (stably overexpressing H19), using antibodies to H3K4me3 and H3K27me3 (Fig. 7).

Differential binding analysis carried out using biological triplicates (n=3) demonstrated enrichment of H3K4me3 marks for NE genes and loss of H3K4me3 marks for AR signaling genes with most of the changes occurring in the proximal promoter regions of these genes (new Fig. 7A-B). For example, V16D/H19 cells showed significant differential binding of H3K4me3 for some NE genes (SOGA3, CDH2, KCNB2, BRINP1) and loss of H3K27me3 binding on other NE genes (ONECUT1, FGF9, MYT1, RUNX1T1, HOXD10) (new Fig. 7D-E). Some NE genes such as REST that are suppressed during NEtD are shown to have a loss of H3K4me3 with H19 overexpression. Dual regulation by H3K4me3[↑]/H3K27me3[↓] are also observed in NE genes such as BRN2, RGS7, ETV5 (new Fig. 7D-E). Together these new findings (page 15 of the revised manuscript) along with the eRRBS data presented in the original submission demonstrating alterations of CpG island methylation on NE genes (new Fig. 8) confirmed that H19 regulates the NEtD by altering the epigenetic landscape of NE genes.

As mentioned above, new data in C4-2B cells showed that androgen depletion elevated SOX2 levels, and this transcription factor drove increases in H19 levels (new Supplementary Fig. 12E). Analysis of SOX2 binding regions in the proximal H19 promoter demonstrated several potential binding sites (new Fig. 5I, top). Examining C4-2B cells treated with Enzalutamide (10 μ M, 6days), ChIP with anti-SOX2 antibody showed enrichment of SOX2 binding on 2 specific sites as compared to IgG (new Fig. 5I, bottom) control. Mutating one of the SOX2 sites on the H19 promoter (new Fig. 5J, top) caused a significant reduction in ENZA-induced H19 promoter luciferase activity (new Fig. 5J, bottom). Together these results (page 14) indicate the transcription of H19 appears to be regulated in part by the binding of SOX2 whose level is markedly increased during androgen deprivation and consequent NE transdifferentiation.

8. Regarding H19 and EZH2/PRC2: It seems that MANY lncRNAs including H19 would bind the EZH2/PRC2 complex but the binding specificity remains obscure in most cases. In the current context, although authors provided nice evidence for the high preservation of the H19 primary sequence as well as secondary structures across the species (Figure 2 and Supplementary figure S3-S5), there is no information on which part of H19 actually binds to which domain(s) of EZH2. Along this line, it's unclear exactly how H19-EZH2 interactions lead to a transition to the NE phenotype?

We have demonstrated that full length H19 bound to EZH2 while Luo and colleagues (PMID 23354591) have shown that the 1062-nt region at the 5'-end is sufficient for this interaction. In contrast to full length H19, the DOX-inducible expression of an EZH2 binding site deletion fragment (H19^{DEL} fragment) (1259-2437 nt) in LNCaP and C4-2B cells, that is unable to bind to

EZH2, does not increase NE markers mRNA and protein expression (new Fig. 3F,6F, Supplementary Fig. 13B-D), suggesting that the 5' terminus is a key driver of these changes.

We presented data in the response to the query to suggest that H19 functioned as an epigenetic modifier of both histone and DNA methylation. This work is detailed in part in our new Fig. 7 and included on page 14-16 of the revised manuscript, in which we review CHIP-seq data derived from cells that overexpress H19.

About expression levels of H19 from RNA-seq and RT-PCR: To compare the overall relative expression levels of H19 in different cohorts and samples/models/contexts, authors should consider presenting the original Ct values of H19 and the normalized read counts or TPM (if possible) in different models (including clinical samples, cell lines, mouse models and organoids) in a Supplementary table.

We have presented the original Ct values and FKPM values for cohorts in new supplementary table 18 and 2 respectively.

9. About box plots: In all box plots throughout the paper, authors should present the n (patient/case numbers) and other parameters such as centerlines and whiskers.

All box plots of manuscript have been regenerated to include these elements. For clarity with our heavy-content figures we intentionally removed patient numbers from these plots. However, we agree that this is important in assessing the data. Therefore, we have promoted our Supplementary Table 1 to a main manuscript (new Table 1) so readers can readily interpret all clinical data presented in Figures 1, 2, and 9. Collectively, these rare patient specimens highlight the significant presence of H19 in NEPC and NEtD.

10. About H19 and alteration in DNA methylation:

- a. As shown in Fig. 3, knocking down of H19 turns off NE genes and turns on luminal genes. What are the changes of methylation status on luminal markers such as CK8, CK18 and NKX3.1 and NE markers.

Although CK8, CK18 and NKX3.1 did not show significance and were not present in the list generated, some luminal and NE markers (RGS7, SPDEF, DLL3 among others) have an altered methylation status. These results can be found in Figure 8 and Supplementary table 15.

- b. The genes listed in Fig. 6C are not consistent with those in Fig. 6D and Fig. 6E. For example, NKX2.1 in Fig. 6D is not shown in Fig. 6C, while BCL11B and CCND2 in Fig. 6E were not presented in Fig. 6D and 6C.

We have revised older Fig. 6 (new Fig. 8) according to the suggestions made by the reviewer.

11. Fig. S1 (top): Is it VPC-P or VPC-M? In this table, p-values should be denoted with * $p < 0.05$, ** $p < 0.01$, and *** $p < 0.001$. As presented, the data can be confusing and misleading. For instance, SOX11 does not seem to differ in the WCDT, SOX9 and many other genes exhibit divergent and opposite changes but authors only vaguely stated “.....(Supplementary figure S1, p-value < 0.05 unless denoted by *).” (page 7, end of the top paragraph). Importantly, INCONSISTENT patterns (ups vs. downs’ of most genes analyzed across the 5 datasets should be discussed.

We apologize for the poor figure labeling and lack of explanation regarding inconsistent patterns. We have regenerated supplementary figure 1 and 3 to respond to comments. In addition, we have added a 3rd supplementary figure to highlight this data. (new Supplementary Figures 1-3). In response to reviewer comment #11 supplementary table 1 has been moved to a main table (Table 1) to allow easier interpretation of this data.

MINOR POINTS:

1. Ref. 28 and 40 are the same.

This has been corrected.

2. Supplementary figure S2, as presented, is not very informative, as there are multiple ‘neuron’, ‘neural’, ‘neurogenesis’ gene expression terms that authors should interrogate. Also, the NES, FDR, and p-value should be presented in the GSEA plot.

Supplementary figure S2 data has been removed. Additional computational and experimental data verify that H19 is essential for NEtD.

3. Fig. 1D has 38 genes instead of 37 genes mentioned in the Text (page 7; second to last line).

This has been corrected to 38 genes.

4. In Supplementary figure S8E, what’s the unit of H19 expression? The Y-axis presentation should start from “0”. In figure S8F, the “+” should be “-” in the absence/presence of DHT.

This has been corrected for Supplementary Figure S12B in the revised manuscript.

5. Supplementary figure S3: n and p-values should be indicated directly in the box plots. The Y-axis unit should be indicated.

Corrected and please see answer to comment #9 and #11 above.

6. How are the sensitivity score and specificity score calculated?

Thank you for noting the exclusion. This has now been included in the supplementary method section.

7. Page 9, third line from the bottom of the top paragraph: 'Figure 4D-H10i....' should be 'Figure 2D....'.

This has been corrected.

8. Figure 6B and Figure 6C were mis-cited in the text (page 13) which did not match the actual data.

As suggested, the mis-citations have been corrected for Figure 8B and 8C in the revised manuscript.

9. There are also many other typographical and grammatical errors.

We have modified the manuscript to correct typographical and grammatical errors.

Reviewer #2 (Remarks to the Author):

This manuscript concerns the functional role of the lncRNA H19 in lineage plasticity and NEPC differentiation. H19 expression increases in treatment induced NEPC relative to CRPC (which is AR dependent) and also may be higher in primary NEPC relative to adenocarcinoma. Using various model systems, the authors also suggest that H19 regulates NE and stem cell genes as well as proliferation and invasion. H19 has previously been found to bind EZH2, which the authors demonstrate occurs in prostate cancer cells and also show here that exogenously modulated H19 is correlated with inverse changes in total H3K27me3 levels. In addition, depletion of H19 in a NEPC cell line resulted in a changed DNA methylation pattern that partially overlapped (~10%) with NEPC/CRPC dichotomy found in clinical samples.

The concept that H19 is a functional component of epigenetic programming associated with adenocarcinoma to neuroendocrine commitment is of interest. At this time, I would consider the data quite preliminary in validating the hypothesis. The analysis of the model systems is incomplete and misinformed in a few instances. Although the mechanistic analyses are a good start, they lack the depth and completeness necessary to draw conclusions concerning mechanisms of action. Finally, the suggestion that H19 may be diagnostically useful relies on a limited number of descriptive cases. A more complete discussion follows.

Figure 1 and bioinformatics analyses: H19 expression appears clearly higher in NEPC than adenocarcinoma (AD). Of particular interest would be how H19 expression varies across CRPC phenotypes. In the WCM-2 cohort, for example, it would be of interest to analyze the available AR signature scores and NE signature scores vs. H19 expression across the CRPC cohort. This type of analysis would be more informative for deciphering a general role in dedifferentiation/lineage plasticity/NE commitment than comparing primary adenocarcinoma to treatment induced NEPC (e.g. Figure 1C). The procedure used for the selection of samples in Figure 1D is not clear as AD samples from some cohorts (e.g. WCM-1 and WCDT) appear to have been left out of the analysis.

As the reviewer suggested, we have interrogated the AR and NEPC signature scores published from the WCM2 cohort (PMID 26855148). We performed correlation analysis with H19 expression in the WCM1 cohort to AR and NEPC signature scores. We observed a significant and strong negative correlation with AR and positive correlation with NEPC (new Fig. 1D). While this result is revealing, WCM1 is composed of clinical AdPC and tNEPC and does not include CRPC samples. Therefore, to further address the reviewer's inquiry and identify the role of H19 in cell

phase commitment, we repeated this analysis on WCM2 (includes CRPC and tNEPC samples) and WCM1 collectively (new Supplementary Fig. S4). We observed a similarly significant negative correlation with the AR signature and positive correlation of H19 expression with NEPC signature scores, respectively. The results (new Fig. 5A, page 13 of the revised manuscript) strongly suggested not only a role in NEPC initiation but also NEPC maintenance – a query brought forward by Reviewer #1 in Question #1. Lastly, we apologize for the typo in the initial (Fig. 1E) unsupervised hierarchical clustering that made the reviewer believe samples were left out of this analysis. All AD samples from WCM1 and VPC cohorts were used and all NEPC samples from VPC, WCM1, WCM2, WCDT, and BCCA were examined. The only patient samples not included were those from the BCCA cohort (matched NEPC with benign samples) which displayed a high level of tumor cellularity (PMID 26855148).

Figures 3: The data in these figures reflect either ectopic expression or knockdown of H19 in a variety of NEPC cell lines derived from patients, experimental human cell lines, or primary human prostate or mouse model organoids. Organoids can be quite sensitive to off-target effects of shRNAs, and therefore additional shH19 constructs should be compared or preferably, rescue experiments should be performed. Most often, relative RNA expression for a few specific genes was compared within the total population. Such data is not clearly interpretable because the actual level of RNA expression is not indicated and could represent a very modest fraction of cells or a physiologically insignificant level of expression. Much more informative and convincing would be histological staining of relevant markers in fixed cell populations or molecular single cell analyses. Organoids from primary tissues are highly heterogeneous, and the methodology of deletion used here did not distinguish the various cell types., other investigators do not find significant histological/molecular NE differentiation for in For example vitro deleted Tp53/Rb1 mouse prostate organoids (Fig. 1J) or in Tp53/Pten null mouse models (Fig S6F). Also, the predominant growth of normal basal organoids from primary human prostate cancer prostatectomy specimens has been observed by many researchers. Therefore, it is incumbent upon the investigators here to demonstrate that Hu-AdPC organoids (Figs. 3E and 4G) are in fact luminal prostate cancer.

We would like to clarify that the methodology used here reduces the possibility of toxic lentiviral transduction effects. The Trp53/Rb1 floxed mouse model was selected to reduce the number of vector constructs needed to achieve the p53/Rb/H19 knockdown. The transduced organoids were passaged at least twice before the cell growth assays were performed. Scrambled shRNA lentivirus as a control were used at the same titer as the shH19 virus. These have been included in the Supplementary Methods section.

Our observation that of the 4 different shRNAs H19 tested only 2 worked (new Supplementary Fig. S8H, Methods) suggested that the anti-proliferative effect of the knockdown was specific to the shRNA and not a toxicity artefact. Using these two shRNAs in 42D cells human H19 was knocked down and reduced the NE phenotype. Rescue experiments were also performed to support these observations. Overexpression of mouse H19 in these cells rescued the neuroendocrine (NE) gene expression suggesting that H19 levels play a critical role in controlling the NE phenotype (new Supplementary Fig. 9I-J). No lentiviral toxic effects were seen.

Unfortunately, in murine Trp53/Rb1 organoids only one shRNA specific for mouse H19 was functional. Additionally, we tested inducible shRNA systems which did not provide us with a measurable knockdown of H19.

We agree with the reviewer that our knockdown of H19 is not targeted to a specific cell population in the organoids. As suggested, IHC staining of the murine organoids (new Fig. 5L) was performed. Results indicated that control organoid cultures grow slowly in predominant luminal glandular structure (A) with strong expression (arrows) of luminal marker (Ck8). Trp53/Rb1 DKO organoids showed solid not cystic growth pattern with abundant cells (arrows) positive for Ki67 and weak Ck8 staining. Inhibition of H19 by shH19 moved the organoids back to a predominantly luminal glandular structure with fewer cells (arrows) positive for Ki67 and strong expression (arrows) of Ck8. *In vitro*, similar to what has been reported (PMID 28059768), we did not observe any NE marker staining in mouse organoids that were knocked out for Trp53 and Rb1 (data not shown).

We concur with the reviewer that the patient-derived organoids are likely basal with less than 1% luminal population. Because of the Covid-19 pandemic we were not able to do further experiments on freshly derived de-identified human prostate cancer tumors. We have hence removed Figure 3E of the original manuscript since we were unable to collect enough samples to validate the luminal/basal phenotype with IHC finding in patient derived organoid cultures at the time.

Figure 4: This figure addresses growth regulation by H19 depletion in NEPC models as well as the suggestion that H19 “re-sensitizes” NE models to androgen antagonism. The comments above about the appropriate characterization of the Tp53/Rb1 null mouse model system and the specificity of shH19 methods apply here. Organoids are quite sensitive to non-specific toxic effects of lentivirus infection. Titrating lentivirus relative to growth inhibition and protein expression is necessary for conclusive organoid growth assays (see Fig 4A). Relevant to growth inhibition, in Fig. 4E, it appears that ENZA treatment and shH19 may be simply additive and not suggestive

of a gain in ENZA sensitivity. The concept that H19 loss leads to AR dependence is an interesting and important one that requires determining AR protein expression levels as well as analyzing multiple AR dependent genes.

As in the prior response, additional experiments did not point to lentiviral toxicity. The viral titers used for the infections are now shown in the Supplementary Methods section.

We neither found that H19 levels regulated AR protein levels nor the half-life of the AR protein (see figure 2 in response to reviewer 1). This result suggested other specific regulation. Our observation that H19 regulated the levels of H3K27me3 and H3K4me3, suggested that potential for epigenetic changes. To examine the role of H19 in regulation of AR signaling, we have performed additional experiments including ChIP-seq (new Fig. 7). ChIP-seq done in V16D cells overexpressing H19 when compared to control V16D cells showed that when H19 is overexpressed, the H3K4me3 active marks in genes that are regulated by AR are reduced. We found that there were distinct changes in both H3K27me3/H3K4me3 in the promoter of androgen responsive genes KLK2, KLK4, NKX3.1 (new Fig. 7D-E). These results (page 15 of the revised manuscript) suggested that H19 epigenetically regulated AR signaling.

To examine how H19 knockdown contributed to sensitivity to ENZA in LNCaP-shTp53/Rb cells- we performed western blots to measure the protein expression levels for AR and NE markers in lysates collected at 48h. Results (new Figure 4I) demonstrated that H19 knockdown suppresses the ENZA mediated upregulation of AR and NE genes. Quantification of these western blots revealed similar results with 2 different shRNAs used for H19 knockdown. These finding have been included on page 13 of the revised manuscript.

Figures 5 and 6: The modulation of total H3K27Me3 relative to H19 levels and the association of H19 with EZH2 and SUZ12 does not show H19-mediated regulation of the PRC2 complex and epigenetic regulation of NEPC genes. This requires specific analyses of H19/PRC2 occupancy and function at selected gene regulatory regions. In addition, DNA methylation studies are at best a highly indirect measure of PRC2 activity.

To examine how H19 interaction with EZH2 was affecting gene transcription, ChiP seq analysis with antibodies to H3K27me3 and H3K4me3 in V16D cells having H19 overexpression was compared to changes in control vector transduced cells (new Fig. 7). Results indicate that differential binding/enrichment of H3K27me3 was associated with transcriptional repression and H3K4me3 was associated with transcriptional activation on NE and AR signaling genes (new Fig. 7). The results of this experiment are further detailed in page 15 of the revised manuscript. This

experiment suggested that H19 binding to EZH2 has profound effects on epigenetic mechanisms regulating the NEPC phenotype.

Reviewer #3 (Remarks to the Author):

In their manuscript titled, “The long noncoding RNA H19 regulates tumor plasticity in neuroendocrine prostate cancer,” Sigh N, Ramnarine VR and colleagues suggest a previously unrecognized role for H19 in regulation of the prostate cancer neuroendocrine phenotype. The work is interesting and contains many novel elements, and the manuscript is well-written. I have the following comments:

1) The well-known function of H19 is in the regulation of the growth factor IGF2. H19 is often silenced through loss of imprinting in human cancers, thereby relieving the suppressive effect on IGF2 expression. Demethylating agents are known to induce re-expression of H19, suppress IGF2 function, and thereby inhibit cancer growth. This well-known function is not explored in the manuscript, and it is difficult to contextualize the findings without an exploration of this well-established H19 signaling axis.

By using qPCR in normal, AdPC and NEPC cell lines and organoids we examined the IGF2 expression. Results indicated a slight increase in IGF2 mRNA from AdPC to NEPC samples (new Supplementary Fig. S14A). These increases are small when compared to the extent of H19 RNA changes (new Fig. 3A). Additionally, methylation analysis of the IGF2/H19 ICR demonstrated increased methylation in prostate cancer cells but methylation was reduced to the normal imprinting range (40-50%) in an NEPC cell line and organoid (new Fig. 3C) (page 10 of the revised manuscript). As noted, IGF2 is known to play a role in driving cancer growth and is regulated by SOX2 (PMID 32427884). Dissecting the role of this protein in NEPC versus prostate cancer growth is complex. Further experiments will be needed to elucidate this relationship.

2) Minor comment: It is unclear what the authors mean by the term “clinically recurrent” (Page 8) in the context of dysregulated gene expression.

Further clarification and explanation for this statement has been provide in our revised manuscript (page 9).

3) The conservation analyses in Figure 2 are interesting, but not entirely novel, and unclear how they support the specific message in the paper. The fact that H19 is a highly conserved noncoding RNA has been known in the field for a long time (see e.g. Smits G et al, Nat Genet, 2008). Conservation of the secondary structure of H19 has also been well-established.

We agree with the comments of the reviewer. Our sequence conservation analysis provides an expansion on these efforts and supports an increased range of biological importance across all

Eutherian species – or at least those that have been sequenced to date. Original reports only included a handful of mostly primate sequences. As mentioned in our original submission, secondary structure conservation (of non-coding RNAs) is suggestive of the functional importance since this is the form of the molecule that carries out its actions. Our methodology for this analysis is novel and we furthermore, to the best of our knowledge, are unaware of any other studies that have shown conservation of H19 secondary structure.

We feel that while this figure does not easily flow with the message of the manuscript, it does point to the conserved importance of H19. If the reviewer disagrees, we are happy to remove Figure 2 and the accompanying supplementary material to allow for publication of this manuscript.

We apologize these points were not clear in our original submission and have made changes in our revised manuscript to rectify that.

4) It is unclear why the authors chose not to consider H19ii as a relevant isoform in NEPC. They only state that it is not reported within GENCODE or RefSeq – but this does not mean it is not relevant.

We apologize for not explaining the rationale for the removal of H19ii more thoroughly. We neglected to mention that this isoform was also removed due to it being a computational artifact related to our sequence analysis pipeline. Regardless, due to the evolving landscape of H19 isoform annotations (it is an unstable transcript) and the time from our original submission to our revised submission, we reprocessed all of our sequence analysis related to isoform identification and quantification to include more updated annotation. We also now include new metrics for how we ranked isoforms (see Supplementary Materials and Methods) to further address the reviewer's important comments. Please see Fig. 2D-E and accompanying Supplementary Figures for our newly generated data (page 9-10 of the revised manuscript).

5) Minor comment: I may have missed it, but what controls were used to assess whether the RNA immunoprecipitation experiments did not simply detect binding of polycomb complex members to the genomic locus encoding H19?

IP lysate prepared from cells (2×10^7 cell equivalents per IP) were subjected to immunoprecipitation using 2 μ g of either normal mouse IgG or a normal rabbit IgG as controls. Successful immunoprecipitation of EZH2 or SUZ12-associated RNA was verified by

qRT-PCR using H19 primers with various negative controls including U1snRNA, IGF2/H19 ICR, or Sox2 for validation of on-target and non-target associated RNA. We have included this part in the Supplementary Methods section of the revised manuscript.

REVIEWER COMMENTS

Reviewer #1 (Remarks to the Author):

This is a significantly improved manuscript and the authors have made conscientious efforts to address most of my previous questions and concerns. They have offered convincing evidence that the lncRNA H19 is upregulated in NEPC and plays a causal role in mediating treatment-induced NE differentiation. They further provide some evidence that H19 interacts with EZH2/PRC2 complex and regulates global methylation. Finally, the authors suggest H19 expression levels as a potential predictive and prognostic biomarker for NEPC. Overall, this is a significant and impactful piece of work that greatly advances our understanding of how an imprinted lncRNA may become dysregulated to modulate PCa cell plasticity and drive PCa aggressiveness.

MAJOR POINTS:

1. With the cornucopia of elegant but multi-faceted data in this project, I recommend the authors, towards the end of their presentation, to develop Figure 5K into a schematic model to summarize the hypothesized MOA of H19 in the context of AR/stemness factor signaling and regulating PCa cell plasticity and NEPC development.
2. To validate their LNCaP-shP53/RB1 cell studies and strengthen the conclusion that loss of H19 induces NE to luminal lineage switching, authors may analyze the levels of luminal markers like CK8/AR/PSA and NE markers such as CHRMA/Syp/Nse along with EZH2 in H19 knockdown LNCaP-WT and LNCaP-shP53/RB1 cells.
3. Figure 6: Authors should analyze EZH2 levels in H19-overexpressing PCa cells (Figure 6C). Also, control group is missing for EZH2 ChIP experiment performed in LNCaP cells (Figure 6E).

MINOR POINTS:

1. In Figure 3F, the Y axis should be "relative RNA expression" but not "relative H19 expression". Please include the cell line information in the figure panel.
2. In Figure 4H, authors showed that ENZA significantly inhibited the growth of shScr infected LNCaP-shP53/RB1 cells, but P53/RB1 double knockdown has been reported earlier to render LNCaP cells completely unresponsive to AR-targeted therapies (Mu et al., Science. 355:84; Figure 1B). Please clarify.
3. Data related to Figure 8C was not properly cited and discussed in the Text (page 16; third paragraph) and there is a typo in the title of Figure 8C in the figure. Also, for Figure 8C, please indicate the total number of genes in each dataset that led to the indicated overlapped genes.
4. Supplementary figure S9H was cited pretty late in the text; so it was unclear which of the shH19 vectors, C or D, was used in some knockdown experiments such as in Figure 3 and 4, for example.
5. Some of the data presented, e.g., Figure S9I-J and Figure S13A, need statistical treatment to boost the conclusions drawn in the text.
6. There are still many typographical and grammatical errors and grantsmanship issues throughout the Text. Below are just some examples.
 - On page 10, both the ending sentence in the first paragraph AND the starting sentence in the last paragraph do not make any sense.
 - On page 11, in the second to last paragraph the 'Supplementary figure S10' was mis-quoted as figure S10 has nothing to do with the Dox-inducible system.
 - Oftentimes, Figures and Figure panels were not presented in sequential order (i.e., presented out of order).

Reviewer #2 (Remarks to the Author):

The revised manuscript is more focused and interpretable. I think it will be of interest to various fields in the scientific community. How directly H19 influences epigenetic regulation remains an important question. The data presented is correlative. Epigenetic changes to lineage loci are expected markers of NE transitions, and changes to lineage marker expression are correlated with H19 expression levels. Whether and possibly how H19 directly influences epigenetic mechanisms through binding to the PRC2 complex is an interesting area for future research.

Reviewer #3 (Remarks to the Author):

The authors have addressed my concerns adequately.

Response to reviewers' comments:

We would like to thank the reviewers for their constructive feedback. We have answered the critiques from reviewer including experimental evidence, biological replicates to improve statistical significance, and significant revision in grammar and syntax within the manuscript. Additionally, we have also revised the manuscript to accommodate editorials requests including reformatting of graphs. We hope that the manuscript in the current form is acceptable for publication.

Please see the response in **RED**.

Reviewer #1 (Remarks to the Author):

This is a significantly improved manuscript and the authors have made conscientious efforts to address most of my previous questions and concerns. They have offered convincing evidence that the lncRNA H19 is upregulated in NEPC and plays a causal role in mediating treatment-induced NE differentiation. They further provide some evidence that H19 interacts with EZH2/PRC2 complex and regulates global methylation. Finally, the authors suggest H19 expression levels as a potential predictive and prognostic biomarker for NEPC. Overall, this is a significant and impactful piece of work that greatly advances our understanding of how an imprinted lncRNA may become dysregulated to modulate PCa cell plasticity and drive PCa aggressiveness.

MAJOR POINTS:

1. With the cornucopia of elegant but multi-faceted data in this project, I recommend the authors, towards the end of their presentation, to develop Figure 5K into a schematic model to summarize the hypothesized MOA of H19 in the context of AR/stemness factor signaling and regulating PCa cell plasticity and NEPC development.

Added as Figure 9D.

2. To validate their LNCaP-shP53/RB1 cell studies and strengthen the conclusion that loss of H19 induces NE to luminal lineage switching, authors may analyze the levels of luminal markers like CK8/AR/PSA and NE markers such as CHRMA/Syp/Nse along with EZH2 in H19 knockdown LNCaP-WT and LNCaP-shP53/RB1 cells.

The lineage switch phenomena is challenging to prove phenotypically in two dimensional cell lines such as LNCaP. Below we show a representative result from several experiments with varying assay conditions (Figure R1) in response to this query. This particular experiment was done in LNCaP cells. We do not feel that two-dimensional cell culture is the best way to see the lineage switch. Due to these issues, we have decided not to add the Response Figure R1 to the manuscript. Note, additionally, the reason to select mouse organoid models was that they served

as better models to depict this switch phenotypically. We have consistently seen the lineage switch with H19 knockdown in P53/RB knockout in organoid models (Fig. 3J and 3K) as well as NEPC cell lines (Fig. 3H, 3I, 3M) to strengthen the conclusion.

Response figure R1: Expression of luminal and basal markers in Wild type LNCaP cells (WT) with P53/RB knockdown (shP53/RB). Two different H19 shRNAs were used for knockdown- shH19-C and shH19-D. Values below the blots indicate mean intensity of the protein band normalized to the respective ACTIN control.

3. Figure 6: Authors should analyze EZH2 levels in H19-overexpressing PCa cells (Figure 6C). Also, control group is missing for EZH2 ChIP experiment performed in LNCaP cells (Figure 6E).

Regarding Figure 6C comment: We have noted (page 14) in the manuscript that **“H19 overexpression in LNCaP was shown not to alter the expression levels of PRC2 complex proteins, EZH2, SUZ12, and AEBP5 (data not shown), whereas RNA immunoprecipitation (RIP) in LASCPC-01 (Figure 6E) and NCI-H660 (Supplementary figure S13A) demonstrated the binding of EZH2 to H19.”**

Below (Response Figure R2) we demonstrate that this is true using overexpression of H19 in V16D cells.

Response figure R2: EZH2 levels in V16D cells with H19 overexpression.

Regarding Figure 6E comment: We were asked to show data for the control group that is missing for EZH2 CHIP experiment performed in LNCaP cells in Figure 6E. In response, we have measured the endogenous interaction between EZH2 and H19 in **control LNCaP** cells (Response Figure R3). LNCaP cells have a very low level of H19. As a result, we were not able to see significant H19 and EZH2 binding due to low endogenous expression of H19 in parental LNCaP cells (See Response figure R3), whereas significant binding enrichment with EZH2 was observed in LNCaP cells overexpressing H19 (Response Figure R3).

Response figure R3: RNA immunoprecipitation to assess EZH2 binding in LNCaP-C (control) and LNCaP-H19 cells. Data are mean +/- SD, n=3, ns depicts a statistically nonsignificant result and * represents p values<0.001**

The rationale behind the left panel of Figure 6E in the present manuscript was meant to demonstrate similar level of H19-EZH2 interaction as in the NE-like cell line LASCPC-01 with LNCaP cells with H19 overexpression. In the middle and the left panel of Figure 6E in the manuscript, we do show the H19-EZH2 and H19-SUZ12 interaction in both the control as well as H19 overexpression in V16D cells.

MINOR POINTS:

1. In Figure 3F, the Y axis should be “relative RNA expression” but not “relative H19 expression”. Please include the cell line information in the figure panel.

Corrected Fig. 3F to demonstrate the relative normalized expression and cell line.

2. In Figure 4H, authors showed that ENZA significantly inhibited the growth of shScr infected LNCaP-shP53/RB1 cells, but P53/RB1 double knockdown has been reported earlier to render LNCaP cells completely unresponsive to AR-targeted therapies (Mu et al., Science. 355:84; Figure 1B). Please clarify.

The reason for the difference between the present data and data from Mu P. et al could be due to the difference of cell preparation.

First, Mu P. et al., utilized the LNCaP/AR with stable AR overexpression, whereas we have used naïve LNCaP for our ENZA studies. Mechanisms related to ENZA resistance have been identified, including AR gene amplification (PMID: 30033370, PMID: 29909985). Thus, high expression of AR would be more susceptible to acquire resistance to ENZA.

Second, our double knockdown LNCaP cells are established by serial transductions with three lentiviruses and used the mix population, whereas Mu P. et al selected the double transduced cells by FACS sorting. Remaining single and un-transduced cells could also show the inhibition of ENZA that was observed, accounting for the difference.

3. Data related to Figure 8C was not properly cited and discussed in the Text (page 16; third paragraph) and there is a typo in the title of Figure 8C in the figure. Also, for Figure 8C, please indicate the total number of genes in each dataset that led to the indicated overlapped genes.

We have now included an explanation of results of Figure 8C (page 16).

Figure 8C was modified to correct the typo and included the total no. of genes in the venn diagram.

4. Supplementary figure S9H was cited pretty late in the text; so it was unclear which of the shH19 vectors, C or D, was used in some knockdown experiments such as in Figure 3 and 4, for example.

Please see the Methods section page 6 line 120 “For all experiments, shH19-C was used unless indicated otherwise”.

5. Some of the data presented, e.g., Figure S9I-J and Figure S13A, need statistical treatment to boost the conclusions drawn in the text.

We have included biological replicates to provide statistical significance. Moreover, the use of multiple NEPC cell lines and organoid models also demonstrates the reproducibility of the findings.

6. There are still many typographical and grammatical errors and grantsmanship issues throughout the Text. Below are just some examples.

- On page 10, both the ending sentence in the first paragraph AND the starting sentence in the last paragraph do not make any sense.
- On page 11, in the second to last paragraph the 'Supplementary figure S10' was mis-quoted as figure S10 has nothing to do with the Dox-inducible system.
- Oftentimes, Figures and Figure panels were not presented in sequential order (i.e., presented out of order).

We have checked and corrected all the grammatical errors and grantsman issues throughout the text. We have tried to maintain the numbering sequence as much as possible, to increase readability.

Reviewer #2 (Remarks to the Author):

The revised manuscript is more focused and interpretable. I think it will be of interest to various fields in the scientific community. How directly H19 influences epigenetic regulation remains an important question. The data presented is correlative. Epigenetic changes to lineage loci are expected markers of NE transitions, and changes to lineage marker expression are correlated with H19 expression levels. Whether and possibly how H19 directly influences epigenetic mechanisms through binding to the PRC2 complex is an interesting area for future research.

We thank the reviewer for the positive feedback.

Reviewer #3 (Remarks to the Author):

The authors have addressed my concerns adequately.

We thank the reviewer for the positive feedback.

REVIEWERS' COMMENTS

Reviewer #1 (Remarks to the Author):

The authors have satisfactorily addressed my questions/comments.